# FER-mediated phosphorylation and PIK3R2 recruitment on IRS4 promotes AKT activation and tumorigenesis in ovarian cancer cells

Yanchun Zhang[1†], Xuexue Xiong[1†], Qi Zhu[1], Jiali Zhang[1], Shengmiao Chen[1], Yuetong Wang[1], Jian Cao[2], Li Chen[1], Linjun Hou[1], Xi Zhao[1], Piliang Hao[1], Jian Chen[3], Min Zhuang[1], Dake Li[2]*, Gaofeng Fan[1]*

[1]School of Life Science and Technology, ShanghaiTech University, Shanghai, China; [2]Department of Gynecology, Women's Hospital of Nanjing Medical University, Nanjing Maternity and Child Health Care Hospital, Nanjing, China; [3]ENT Institute and Department of Otorhinolaryngology, Eye & ENT Hospital, Fudan University, Shanghai, China

*For correspondence:
lidake2002@163.com (DL);
fangf@shanghaitech.edu.cn (GF)

†These authors contributed equally to this work

Competing interest: The authors declare that no competing interests exist.

**Abstract** Tyrosine phosphorylation, orchestrated by tyrosine kinases and phosphatases, modulates a multi-layered signaling network in a time- and space-dependent manner. Dysregulation of this post-translational modification is inevitably associated with pathological diseases. Our previous work has demonstrated that non-receptor tyrosine kinase FER is upregulated in ovarian cancer, knocking down which attenuates metastatic phenotypes. However, due to the limited number of known substrates in the ovarian cancer context, the molecular basis for its pro-proliferation activity remains enigmatic. Here, we employed mass spectrometry and biochemical approaches to identify insulin receptor substrate 4 (IRS4) as a novel substrate of FER. FER engaged its kinase domain to associate with the PH and PTB domains of IRS4. Using a proximity-based tagging system in ovarian carcinoma-derived OVCAR-5 cells, we determined that FER-mediated phosphorylation of Tyr779 enables IRS4 to recruit PIK3R2/p85β, the regulatory subunit of PI3K, and activate the PI3K-AKT pathway. Rescuing *IRS4*-null ovarian tumor cells with phosphorylation-defective mutant, but not WT IRS4 delayed ovarian tumor cell proliferation both in vitro and in vivo. Overall, we revealed a kinase-substrate mode between FER and IRS4, and the pharmacological inhibition of FER kinase may be beneficial for ovarian cancer patients with PI3K-AKT hyperactivation.

## Editor's evaluation

This study was designed to examine the role of the FER/IRS4 pathway in ovarian cancer cells. The authors show that FER causes tyrosine phosphorylation of IRS4 and recruitment of PIK3R2 that subsequently causes activation of AKT. The data presented suggest that pharmacological targeting of FER/IRS4 may be beneficial for the treatment of ovarian cancer.

## Introduction

Ovarian cancer is the most devastating gynecological malignancy, with high morbidity and ranking fifth among all cancer-related mortality in women (*Siegel et al., 2020*). Patients with ovarian carcinoma are usually diagnosed at an advanced stage, resulting in a very low 5-year survival rate (*Siegel et al., 2020*). The lack of a genetically engineered mouse model for ovarian cancer significantly delays

the entire process of ovarian cancer research. In particular, the molecular mechanisms of ovarian tumor progression and metastasis are not well understood, key factors severely restricting the overall survival from ovarian cancer (**Binaschi et al., 2011**). Consequently, there is an urgent need and enormous translational potential in revealing the molecular mechanisms regulating the initiation and progression of ovarian cancers. Understanding these mechanisms will serve as the first step toward identifying novel therapeutic targets and biomarkers for intervention against this heterogeneous and deadly disease (**Kulasingam et al., 2010**).

Protein tyrosine kinases represent a family of important enzymes for controlling cell proliferation, motility, survival, and differentiation, whose dysfunction has been closely related to the etiology of many major diseases, including cancer. Depending on the cellular localization, the family can be further divided into receptor tyrosine kinase, which resides in the plasma membrane, and non-receptor tyrosine kinase, which resides in cytosol. The feline sarcoma kinase FES and feline sarcoma-related kinase FER represent a unique family of non-receptor tyrosine kinases. They are characterized by distinguishable N-terminal phospholipid binding and a membrane targeting FER/CIP4 homology/Bin1/amphiphysin/RVS (F-BAR) domain, which are reported to function in cell proliferation, motility, cell-to-cell adhesion, and mediate signal transmission from cell surfaces to the cytoskeleton (**Greer, 2002**; **Craig, 2012**). Notably, the FER protein has been shown to be aberrantly upregulated and activated in different types of carcinoma (**Yang et al., 2009**; **Zoubeidi et al., 2009**; **Albeck and Brugge, 2011**; **Ren et al., 2012**). Specifically, high activity of FER protein kinase has been observed in 22% of ovarian cancer tumor samples via a global phosphoproteomic approach (**Ren et al., 2012**). To date, several receptor tyrosine kinases have been reported to act upstream of FER, including EGFR, PDGFR, and integrin (**Craig, 2012**; **Kim and Wong, 1995**; **Ivanova et al., 2013**). Meanwhile, STAT3, cortactin, MET, and CRMP2, the functions of which are intensively involved in tumor cell motility and chemo-resistance, have been verified as FER substrates (**Orlovsky et al., 2002**; **Fan et al., 2004**; **Zheng et al., 2018**; **Fan et al., 2016**; **Zhang et al., 2018**).

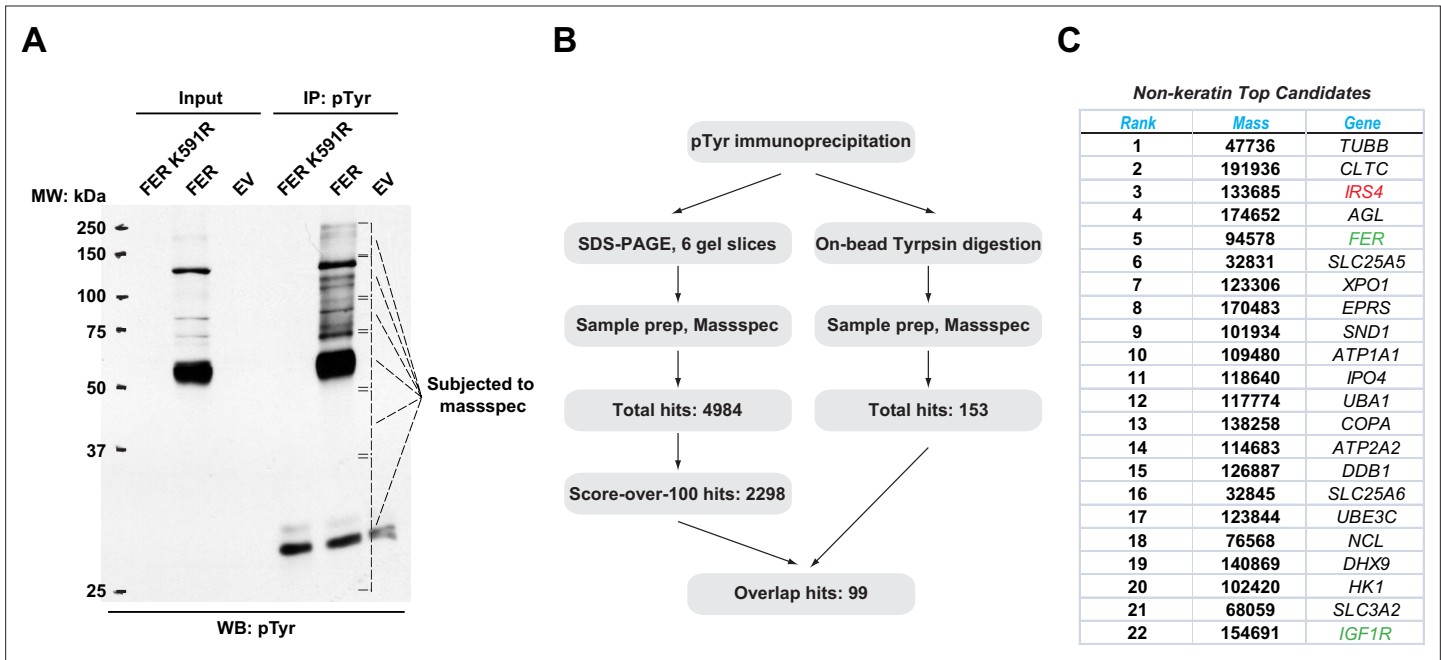

**Figure 1.** Mass spectrometry analysis identified insulin receptor substrate 4 (IRS4) as a novel substrate of FER. (**A**) FER-WT or FER-K591R was transfected into HEK293FT cells, followed by immunoprecipitation with antibody against pTyr. Immunoprecipitates and whole cell lysates were subjected to immublotting analysis with anti-pTyr antibody (**Figure 1—source data 1**). (**B**) Experimental flow chart and results of two methods for potential substrates identification of FER kinase. The immunoprecipitated samples were subjected to in-gel (left) or on-bead (right) tryptic digestion, followed by mass spectrometry protein identification. (**C**) Top-ranked overlapping candidate genes from both mass spectrometry analyses listed in B. Keratin genes were not shown.

The online version of this article includes the following source data for figure 1:

**Source data 1.** Original images for Western blots in **Figure 1A**.

In addition to motility, FER expression has also been associated with proliferation of tumor cells (*Zoubeidi et al., 2009*; *Allard et al., 2000*; *Lennartsson et al., 2013*; *Oneyama et al., 2016*; *Ivanova et al., 2019*). However, due to the limited number of known substrates, the molecular basis for its pro-proliferation activity remains enigmatic. In this current study, we aim to identify novel substrate(s) of FER and investigate the role of kinase-substrate regulatory modules in promoting ovarian tumorigenesis and progression. By integrating mass spectrometry analysis with biochemical and biological approaches, we have demonstrated that FER directly phosphorylates insulin receptor substrate 4 (IRS4) and that this tyrosine phosphorylation is important to create a binding site for recruiting PIK3R2/p85β. As the key regulatory subunit of PI3K kinase, recruitment of PIK3R2/p85β onto IRS4 is required for PI3K-AKT signaling pathway activation and tumorigenesis in ovarian cancer.

## Results

### Mass spectrometry analysis identified IRS4 as a novel substrate of FER

To identify novel substrates for tyrosine kinase FER, tyrosine phosphorylated proteins in HEK293FT cells upon wild-type (WT) and a kinase-dead mutant (K591R) of FER transfection were enriched by the anti-pTyr antibody, 4G10. We observed that significant numbers of proteins could be modified by tyrosine phosphorylation in cells expressing FER-WT compared to FER-K591R (*Figure 1A*). We excised the gel according to the position and prepared these samples via in-gel tryptic digestion for mass spectrometry analysis. We obtained 2298 candidate substrates with scores above 100 (*Figure 1B* and *Supplementary files 1-6*). Concurrently, immunoprecipitated proteins subjected to pTyr (4G10) antibody pull-down were directly digested on beads with trypsin and subjected to mass spectrometry. With this procedure we identified 153 candidate proteins (*Figure 1B* and *Supplementary file 7*). Combining results from the two experiments, we obtained 99 overlapped hits for further examination (*Figure 1B*).

The top 22 candidate genes are illustrated in *Figure 1C*. Notably, we identified FER as its own substrate since the Tyr402 residue of the kinase is known to be auto-phosphorylated (*Fan et al., 2016*). We also identified *IGF1R*, a key tyrosine kinase receptor in regulating cancer cell survival, proliferation, and motility (*Stanicka et al., 2018*). In addition, a number of unreported genes were also listed, including *XPO1* and *IPO4* (cytoplasm-nucleus shuttle; *Azizian and Li, 2020*; *Xu et al., 2019*), *SLC25A5*, *SLC25A6*, and *SLC3A2* (ADP/ATP transportation from mitochondria to cytoplasm, as well as heteromeric amino acid and polyamine transportation; *Clémençon et al., 2013*; *Palacín and Kanai, 2004*), and *UBA1* and *UBE3C* (ubiquitin-proteasome degradation; *Groen and Gillingwater, 2015*; *Kuo and Goldberg, 2017*). In this study, we focused on IRS4 as our top candidate: (1) #3 in score ranking. (2) A signaling molecule in the same pathway (IGF1R) has been previously identified as substrate for FER kinase. (3) Tyrosine phosphorylation regulation is essential for the biological function of IRS4.

### FER engaged its kinase domain to associate with PH and PTB domains of IRS4

We first investigated if there was any physical interaction between FER and IRS4. We transiently expressed Myc-tagged IRS4 alone or together with FER in HEK293FT cells, followed by immunoprecipitation with resin against a Myc-tag. As shown in *Figure 2A–B*, IRS4 binds to FER and this association was not affected when FER-WT was replaced with its kinase-dead mutant FER-K591R. Interestingly, the binding of GRB2 to IRS4 was FER kinase activity-dependent (*Figure 2B*).

Masanori Iwanishi and his colleagues have demonstrated interaction between IRS1 and FER in 3T3-L1 adipocytes (*Iwanishi et al., 2000*). Interestingly, compared to anti-IgG control, the OVCAR-5 cell lysates with anti-FER antibody showed the interaction between FER and IRS1 at endogenous level (*Figure 2—figure supplement 1A*). We further constructed IRS1 and IRS4 plasmids, and expressed these constructs alone or in combination, as indicated, in HEK293FT cells (*Figure 2—figure supplement 1B*). Notably, the binding between FER and IRS4 was not affected in the absence and/or presence of IRS1.

To further map the region(s) involved in the interaction between IRS4 and FER, we employed different truncation forms of IRS4 with a Myc-tag, as indicated in *Figure 2C*. These truncation mutants were overexpressed when combined with FER and the binding between them and FER was compared

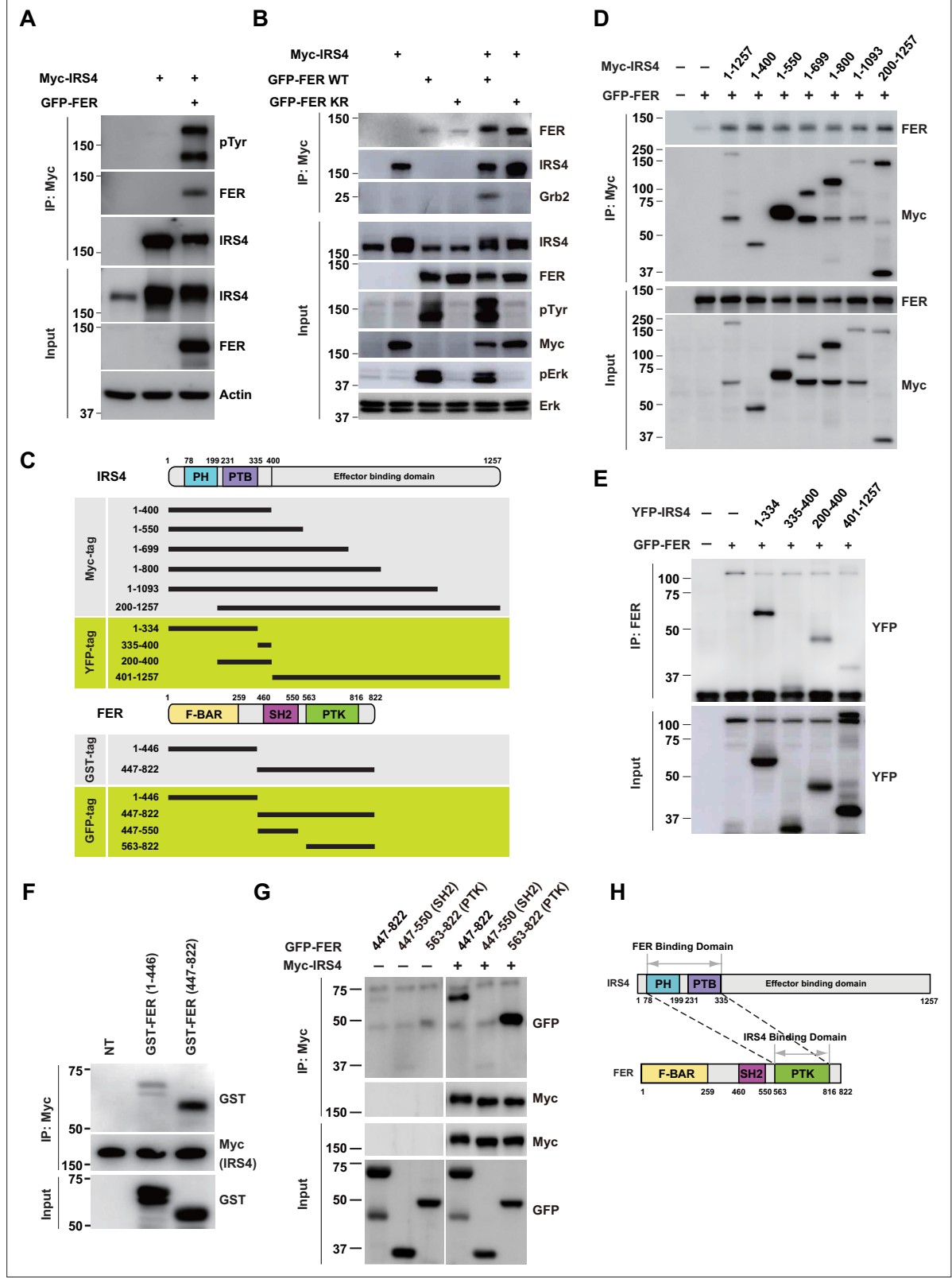

**Figure 2.** FER engaged its kinase domain to physically interact with PH and PTB domains of insulin receptor substrate 4 (IRS4). (**A**) Myc-tagged IRS4 were transiently transfected alone, or together with FER in HEK293FT cells, followed by immunoprecipitation with resin against Myc-tag, and immunoblotting with IRS4, FER, 4G10 antibodies, respectively. The interaction of FER with IRS4 was examined. Expressions of IRS4, FER, and loading control actin in whole cell lysate samples were also probed (*Figure 2—source data 1*). (**B**) After transient transfection with Myc-tagged IRS4, FER-WT,

*Figure 2 continued on next page*

*Figure 2 continued*

or KR mutant in HEK293FT cells, IRS4 was immunoprecipitated with resin against Myc-tag from cell lysates and probed for association with FER and GRB2. The blot was also probed with antibodies against IRS4, FER, 4G10, Myc, phosphor- and total-ERK in whole cell lysate samples (*Figure 2—source data 2*). (**C**) Schematic illustration of domain structure of IRS4 and FER. PH: pleckstrin homology; PTB: phosphotyrosine binding. Multiple truncation mutants of IRS4, including fragments corresponding to amino acid residues 1–400, 1–550, 1–699, 1–800, 1–1093, 200–1257, 1–334, 335–400, 200–400, and 401–1257, with Myc or YFP tag were also listed. F-BAR: FER/CIP4 homology/Bin1/amphiphysin/RVS; SH2: Src homology 2; PTK: protein tyrosine kinase; multiple truncation mutants of FER, including fragments corresponding to amino acid residues 1–446, 447–822, 447–550, and 563–822, with GST or GFP tag were also listed. (**D**) FER was expressed alone or co-expressed with a series of Myc-tagged truncated forms of IRS4 in HEK293FT cells, as indicated. Lysates were harvested and immunoprecipitated with anti-Myc resin. The associations of FER with different IRS4 truncation mutants were compared. Expressions of IRS4 and FER in whole cell lysate samples were also probed (*Figure 2—source data 3*). (**E**) HEK293FT cells were transiently co-transfected with GFP-tagged FER and truncated versions of IRS4 with YFP tag, followed by immunoprecipitation with FER antibody and immunoblotting with YFP antibody (*Figure 2—source data 4*). (**F**) Two truncated FER protein with GST tag were expressed and purified from *Escherichia coli*. Recombinant proteins were further incubated with whole HEK293FT cell lysates expressing Myc-tagged IRS4, respectively. Immunoprecipitates were subjected to immunoblotting analysis to assess the region requirement for IRS4 interaction (*Figure 2—source data 5*). (**G**) IRS4 was expressed alone or co-expressed with a series of GFP-tagged truncated forms of FER in HEK293FT cells, as indicated. Lysates were harvested and immunoprecipitated with anti-Myc resin. The associations of IRS4 with different FER truncation mutants were compared. Expressions of IRS4 and FER in whole cell lysate samples were also probed (*Figure 2—source data 6*). (**H**) Schematic illustration of regions involved in binding between IRS4 and FER.

The online version of this article includes the following source data and figure supplement(s) for figure 2:

**Source data 1.** Original images for Western blots in *Figure 2A*.

**Source data 2.** Original images for Western blots in *Figure 2B*.

**Source data 3.** Original images for Western blots in *Figure 2D*.

**Source data 4.** Original images for Western blots in *Figure 2E*.

**Source data 5.** Original images for Western blots in *Figure 2F*.

**Source data 6.** Original images for Western blots in *Figure 2G*.

**Figure supplement 1.** The association between FER and insulin receptor substrate 4 (IRS4) was not affected in the absence and/or presence of IRS1.

**Figure supplement 1—source data 1.** Original images for Western blots in *Figure 2—figure supplement 1A*.

**Figure supplement 1—source data 2.** Original images for Western blots in *Figure 2—figure supplement 1B*.

**Figure supplement 2.** FER engaged its kinase domain to physically interact with PH and PTB domains of insulin receptor substrate 4 (IRS4).

**Figure supplement 2—source data 1.** Original images for Western blots in *Figure 2—figure supplement 2A*.

**Figure supplement 2—source data 2.** Original images for Western blots in *Figure 2—figure supplement 2B*.

**Figure supplement 2—source data 3.** Original images for Western blots in *Figure 2—figure supplement 2C*.

**Figure supplement 2—source data 4.** Original images for Western blots in *Figure 2—figure supplement 2D*.

**Figure supplement 3.** Further evaluation of the PH and PTB domain of insulin receptor substrate 4 (IRS4) in binding with the kinase domain of FER.

**Figure supplement 3—source data 1.** Original images for Western blots in *Figure 2—figure supplement 3C*.

**Figure supplement 3—source data 2.** Original images for Western blots in *Figure 2—figure supplement 3D*.

**Figure supplement 3—source data 3.** Original images for Western blots in *Figure 2—figure supplement 3E*.

**Figure supplement 3—source data 4.** Original images for Western blots in *Figure 2—figure supplement 3F*.

(*Figure 2D*). Surprisingly, the shortest truncation of IRS4 (1–400) maintained a similar binding affinity with FER, indicating the N-terminal region of IRS4 is crucial for the interaction (*Figure 2D*).

To further pinpoint the key region involved in binding, we employed additional truncated versions of IRS4 with a YFP-tag (*Figure 2C*). Whereas the 335–400 and 401–1257 mutants demonstrated no binding affinity with FER, the 200–400 mutant maintained weak but substantial interaction with FER (*Figure 2E*). The N-terminal mutant 1–334 showed strongest binding among all these truncated constructs (*Figure 2E*). Compared to the 200–400 mutant, which only covers PTB domain, the N-terminal 1–334aa of IRS4 contains both PH and PTB domains. Therefore, we constructed the 1–400 (both PH and PTB domains), 1–200 (PH domain only), and 200–400 (PTB domain only) mutants of IRS4 to further narrow down the binding region on IRS4. However, all three mutants showed as strong binding affinity with FER as WT IRS4, suggesting both PH and PTB domains were involved (*Figure 2—figure supplement 2A*). Meanwhile, we also constructed Myc-IRS4ΔPH and IRS4ΔPTB mutants to further dissect their individual roles in association with FER. Interestingly, deletion of PH or PTB domain has minimal effect on binding affinity with FER (*Figure 2—figure supplement 2B*), suggesting disrupt either of these domains is not sufficient to collapse the protein complex. Together, these results

indicated that both PH and PTB domains of IRS4 are participated in the association with the kinase FER.

Subsequently, we expressed and purified two truncated FER proteins with a GST-tag in *Escherichia coli* (*Figure 2F*) and incubated them separately with whole HEK293FT cell lysates expressing Myc-tagged IRS4 to evaluate their interaction. The C-terminal region (447–822aa, SH2+ kinase domains) of FER, rather than its N-terminal region (1–446aa, F-BAR+ FX domains), showed robust interaction with IRS4 (*Figure 2F*). We further constructed GFP-FER 447–822 (SH2+ kinase domains), GFP-FER 447–550 (SH2 domain), and GFP-FER 563–822 (kinase domain) truncation mutants to narrow down the binding region on FER kinase, as indicated in *Figure 2G*. Consistent to our in vitro purified protein binding results, the 447–822 mutant of FER (SH2+ kinase domains) was important for the interaction with IRS4 (*Figure 2G*). Notably, FER kinase domain, but not SH2 domain, was involved in interaction with IRS4 (*Figure 2G*). Furthermore, FER kinase domain, but not SH2 domain, showed strong interaction with either PH domain (1–200aa) or PTB domain (200–400aa) of IRS4 (*Figure 2—figure supplement 2C-D*).

The pleckstrin homology (PH) domain is a functional domain present in a variety of signaling and cytoskeleton-related proteins (*Maffucci and Falasca, 2001*). The polarity of the PH domain suggests that the ligand may be negatively charged. Therefore, we analyzed the charge distribution on the surface of FER kinase domain. First, we obtain the crystal structure of the target protein from the AlphaFold Protein Structure Database. Then, we used UCSF Chimera v1.14 (https://www.cgl.ucsf.edu/chimera/) to display the 3D structures and label the charged amino acids (*Figure 2—figure supplement 3A-B*). As expected, the surface of the PH domain in IRS4 is mainly distributed with positive charges (*Figure 2—figure supplement 3A*). Interestingly, there is a negative charge distribution on the surface of FER kinase domain, where E676, D684, and E740 are key amino acid residues (*Figure 2—figure supplement 3B*). We constructed single or multiple mutants of these key amino acids of FER, and detected their interaction with IRS4 full-length or 1–200 (PH) mutants, respectively, to verify whether the mutations of negative amino acids in the FER kinase domain would affect their binding with IRS4. The results suggested that the single or multiple mutations of negative amino acids in the FER kinase domain failed to disrupt their interaction with WT IRS4 or 1–200 (PH) mutants of IRS4 (*Figure 2—figure supplement 3C-E*).

The previous work with IRS1 suggests that an NPXY motif might be expected as the PTB domain binding site (*Wolf et al., 1995*). FER possesses no NPXY motif but a QPVY motif within its kinase domain. To test the necessity of this motif in binding with IRS4, we either mutated the key tyrosine residue (Y634F) or deleted this motif (Δ631–634) completely. However, compared to FER-WT, these FER mutants showed an equivalent binding affinity with IRS4 (*Figure 2—figure supplement 3F*).

Combining together, our data suggest that both PH and PTB domains of IRS4 participate in the association with the kinase domain of FER (*Figure 2H*).

## Mass spectrometry analysis and site-directed mutagenesis identified several FER-phosphorylated tyrosine residues on IRS4

We next investigated the molecular details in the tyrosine phosphorylation of IRS4 by FER kinase. The first question we wanted to address was the specificity of regulation. To this end, we overexpressed seven non-receptor tyrosine kinases in parallel and assessed the phosphorylation extent change of IRS4 post immunoprecipitation followed by anti-phosphotyrosine blotting analysis. As shown in *Figure 3A*, among all these kinases, FER illustrated highest capability for tyrosine phosphorylation of IRS4.

To address whether IRS4 is a direct FER substrate, we expressed and purified Myc-tagged human IRS4 protein in HEK293FT cells, and set up in vitro kinase (IVK) assay with purified GST-tagged human FER tyrosine kinase domain (541–822aa). The tyrosine phosphorylation level of IRS4 was increased in an FER kinase dosage-dependent manner (*Figure 3B*). This result demonstrates that IRS4 is a bona fide FER substrate.

To determine which region on IRS4 can be phosphorylated by FER, we overexpressed abovementioned Myc-tagged IRS4 truncation mutants in conjunction with FER in HEK293FT cells (*Figure 3C*) and evaluated the phosphorylation extent change of IRS4 fragments. We concluded tyrosine residues between 700 and 1093 amino acids of IRS4 were potential substrate(s) for FER kinase, since: (1) the 1–1093 mutant of IRS4 showed equivalent phosphorylation level to full-length IRS4 upon FER

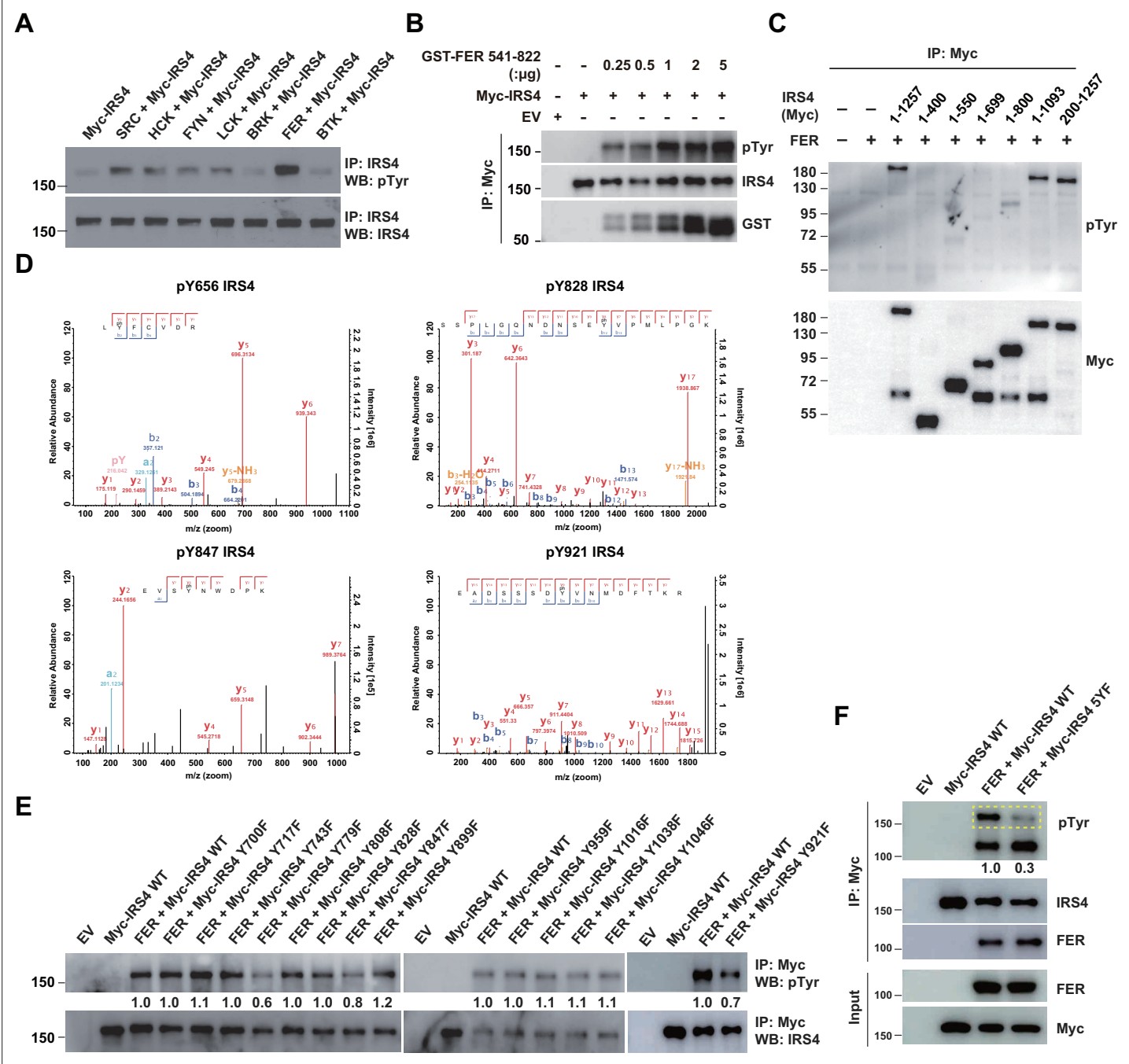

**Figure 3.** Mass spectrometry analysis and site-directed mutagenesis identified several FER-phosphorylated tyrosine residues on insulin receptor substrate 4 (IRS4). (**A**) Myc-IRS4 was transfected alone, or together with several protein-tyrosine kinases in HEK293FT cells, as indicated. Myc-IRS4 was immunoprecipitated with resin against Myc-tag and the phosphorylation level of IRS4 was examined by pTyr (4G10) immunoblotting (***Figure 3—source data 1***). (**B**) Myc-tagged human IRS4 protein was expressed and purified in HEK293FT cells, and in vitro kinase (IVK) assay with purified GST-tagged human FER tyrosine kinase domain (541–822aa) was performed (***Figure 3—source data 2***). (**C**) After transient transfection of a series of Myc-tagged different truncated forms of IRS4 along with FER in HEK293FT cells, as indicated, lysates were harvested and immunoprecipitated with anti-Myc resin. The tyrosine phosphorylation level of IRS4 truncations was examined with antibody against pTyr (4G10) (***Figure 3—source data 3***). (**D**) Results of mass spectrometry analysis identified four tyrosine residues (Y656, Y828, Y847, and Y921) of IRS4 phosphorylated by FER kinase. (**E–F**) HEK293FT cells were transiently transfected with Myc-tagged IRS4 or its YF mutants with FER, as indicated, followed by immunoprecipitation with resin against Myc-tag and immunoblotting with IRS4, FER, pTyr (4G10) antibody. The phosphorylation levels of different IRS4 mutants (quantified by ImageJ software) were compared to illustrate the effects of point mutations in IRS4 on its phosphorylation by FER (***Figure 3—source data 4–5***).

The online version of this article includes the following source data for figure 3:

*Figure 3 continued on next page*

*Figure 3 continued*

**Source data 1.** Original images for Western blots in *Figure 3A*.

**Source data 2.** Original images for Western blots in *Figure 3B*.

**Source data 3.** Original images for Western blots in *Figure 3C*.

**Source data 4.** Original images for Western blots in *Figure 3E*.

**Source data 5.** Original images for Western blots in *Figure 3F*.

phosphorylation; (2) compared to the 1–699 mutant, 1–800 mutant of IRS4 showed weak but significant amount of phosphorylation.

We took two strategies to further pinpoint the tyrosine residue(s) that undergo phosphorylation in the presence of FER. By performing mass spectrometry analysis in duplicate, we observed three tyrosine sites, namely Y656, Y828, and Y921, whose phosphorylation were repeatedly detected in both datasets (*Figure 3D* and *Supplementary files 8-9*). Y847 was also detected once (*Figure 3D* and *Supplementary file 8*). Interestingly, these four residues are within or very close to the 700–1093 region of IRS4. Afterward, we mutated all tyrosine residues within 700–1093 region of IRS4 to phenylalanine and assessed the phosphorylation level change of these mutants compared to WT IRS4. Excluding the Y828, Y847, and Y921 mutants, a tyrosine to phenylalanine substitution at residue 779 remarkably decreased the phosphorylation level of IRS4 in the presence of FER (*Figure 3E*).

By combining results from both mass spectrometry analysis and site-directed mutagenesis analysis, we generated a quintuple Tyr to Phe mutant of IRS4, including Y656, Y779, Y828, Y847, and Y921 (named '5YF' mutant hereafter). The tyrosine phosphorylation level of 5YF mutant by FER was profoundly diminished compared to each single mutant (*Figure 3F*). These results highly suggest that there are five major tyrosine residues within IRS4 that are subjected to FER-mediated phosphorylation.

## IRS4 was upregulated in certain ovarian carcinoma-derived cell lines and important for PI3K-AKT pathway activation and ovarian cancer cell proliferation

Previous studies from our lab have demonstrated the aberrantly high expression of FER kinase in ovarian cancer and its important role in promoting tumor cell metastasis both in vitro and in vivo (*Fan et al., 2016*). In this study we want to adapted the similar ovarian cancer cell model to further evaluate the biological function of the FER-IRS4 kinase-substrate pair in a physiological context. We first used immunoblotting analysis to compare the expression level of the IRS4 protein between 2 human ovarian surface epithelial (HOSE) cell lines immortalized by the human papilloma viral oncogenes E6 and E7, and 14 ovarian carcinoma-derived cell lines. Compared to normal HOSE control cells, two ovarian carcinoma-derived cell lines, OVCAR-5 and OVCAR-3, showed evident upregulation of IRS4 protein expression (*Figure 4A*). Of note, FER is also upregulated in both cell lines (*Fan et al., 2016*).

Next, we stably expressed ectopic IRS4 into those ovarian carcinoma-derived cell lines with no IRS4 expression, such as HEY and OVCAR-8. The effect of overexpression was confirmed via immunoblotting against an anti-IRS4 antibody (*Figure 4B and D*). Notably, Cell Titer-Glo (CTG) luminescent cell viability assay demonstrated that the expression of IRS4 significantly elevated the proliferation capacity of these cells (*Figure 4C and E*).

Meanwhile, we applied the CRISPR-Cas9 system to genetically knock out the *IRS4* gene in those ovarian carcinoma-derived cell lines with high IRS4 expression, such as OVCAR-5 and OVCAR-3. Regrettably, after several attempts we couldn't get a single clone of *IRS4* knockout (KO) OVCAR-3 cells, since this cell line prefers to grow in clusters and it cannot re-confluence after FACS procedure. Nevertheless, we successfully obtained two OVCAR-5 *IRS4* KO cell lines with two distinct sgRNAs (*Figure 4F*).

A series of assays were applied to assess the possible changes in cancer cell fitness in the absence of IRS4. Results from Annexin V-FITC and propidium iodide (PI) double staining assay indicated that loss of IRS4 has minimal effect on OVCAR-5 cell survival (*Figure 4—figure supplement 1A*). Further cell cycle analysis in both WT and *IRS4* KO cell lines with PI staining indicated that loss of IRS4 significantly decreased the proportion of cells within S phase (*Figure 4—figure supplement 1B-C*). Consistently, CTG assay demonstrated that the KO of *IRS4* notably delayed the proliferation capacity of OVCAR-5 cells (*Figure 4G*). Ectopic re-expression of IRS4, rather than an empty vector, restored the

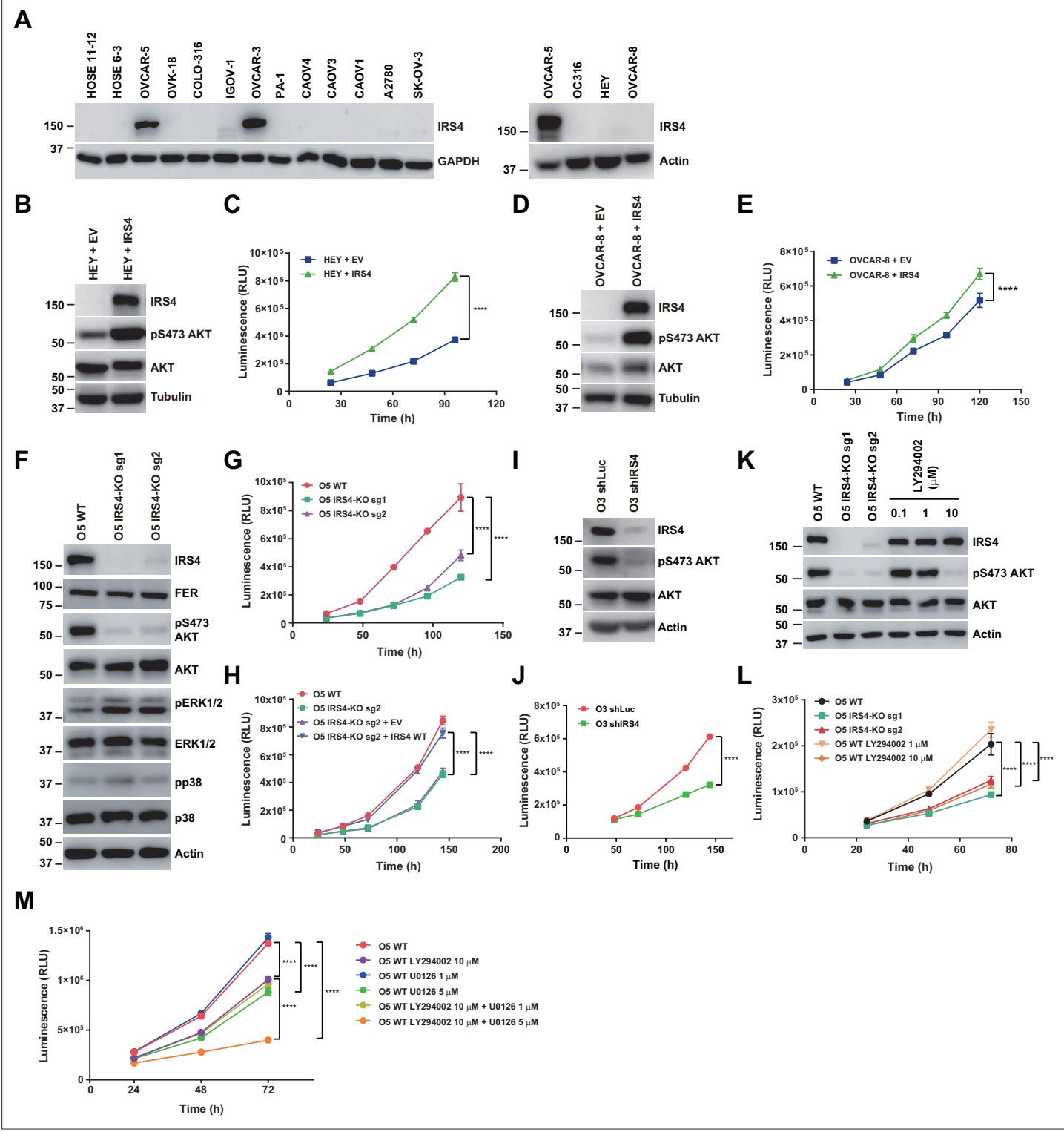

**Figure 4.** Insulin receptor substrate 4 (IRS4) was upregulated in certain ovarian carcinoma-derived cell lines and important for PI3K-AKT pathway activation and ovarian cancer cell proliferation. (**A**) Immunoblotting of IRS4 with two immortalized human ovarian surface epithelial (HOSE) cell lines and 14 ovarian carcinoma-derived cell lines to demonstrate increased expression level of IRS4 in certain ovarian carcinoma-derived cell lines (***Figure 4—source data 1***). (B and D) Immunoblotting analysis to illustrate the overexpression effect of IRS4 on activation of downstream signaling pathways. Sample lysates were prepared from parental and IRS4 ectopically expressed HEY (**B**) and OVCAR-8 (**D**) cell lines, and immunoblotted with the designated antibodies, including IRS4, pS473- and total-AKT antibodies, with tubulin as loading control (***Figure 4—source data 2–3***). (C and

*Figure 4 continued on next page*

*Figure 4 continued*

E) Cell Titer-Glo luminescent cell viability assay with parental and IRS4 ectopically expressed HEY (**C**) and OVCAR-8 (**E**) cells at the indicated time intervals. Results represented means ± SD from six replicates (HEY) or four replicates (OVCAR-8). Significance was determined with two-way ANOVA. ****p < 0.0001. (**F**) Immunoblotting analysis to illustrate the knockout effect of IRS4 on activation of downstream signaling pathways. Sample lysates were prepared from parental and *IRS4*-KO OVCAR-5 cell lines, and immunoblotted with the designated antibodies, including IRS4, pS473- and total-AKT, phosphor- and total-ERK, and phosphor- and total-p38 antibodies, with actin as loading control (*Figure 4—source data 4*). (**G**) CellTiter-Glo luminescent cell viability assay with OVCAR-5 wild-type (WT) cells and OVCAR-5 *IRS4*-KO cells at the indicated time intervals. Results represented means ± SD from six replicates. Significance was determined with two-way ANOVA. ****p < 0.0001. (**H**) CellTiter-Glo cell viability assay was conducted to evaluate cell proliferation after restoration of IRS4 expression in OVCAR-5 *IRS4*-KO cells. The ectopic expression of empty vector (EV) was used as a negative control. Results represented means ± SD from six replicates. Significance was determined with two-way ANOVA. ****p < 0.0001. (**I**) Immunoblotting analysis to illustrate the knockdown effect of *IRS4* on activation of downstream signaling pathways. Sample lysates were prepared from OVCAR-3 shLuc and shIRS4 cells, and immunoblotted with the designated antibodies, including IRS4, pS473- and total-AKT antibodies, with actin as loading control (*Figure 4—source data 5*). (**J**) CellTiter-Glo luminescent cell viability assay with OVCAR-3 shLuc cells and OVCAR-3 shIRS4 cells at the indicated time intervals. Results represented means ± SEM from six replicates. Significance was determined with two-way ANOVA. ****p < 0.0001. (**K**) Lysates from OVCAR-5 WT and *IRS4*-KO cells were harvested and the expression of pS473- and total-AKT was detected by immunoblotting. PI3K inhibitor LY294002 (0.1, 1, and 10 µM), as indicated, were incubated for 24 hr (*Figure 4—source data 6*). (**L**) Cell proliferation was assessed in OVCAR-5 cells following *IRS4* knockout (*IRS4*-KO sg1 or sg2) or incubation with PI3K inhibitor LY294002 (1 and 10 µM) using the CellTiter-Glo luminescent cell viability assay at the indicated time intervals. Results represented means ± SD from six replicates. Significance was determined with two-way ANOVA. ****p < 0.0001. (**M**) Cell proliferation was assessed in OVCAR-5 cells following incubation with PI3K inhibitor LY294002 (10 µM) along with ERK pathway inhibitor U0126 (1 and 5 µM) using the CellTiter-Glo luminescent cell viability assay at the indicated time intervals. Results represented means ± SD from five replicates. Significance was determined with two-way ANOVA. ****p < 0.0001.

The online version of this article includes the following source data and figure supplement(s) for figure 4:

**Source data 1.** Original images for Western blots in *Figure 4A*.

**Source data 2.** Original images for Western blots in *Figure 4B*.

**Source data 3.** Original images for Western blots in *Figure 4D*.

**Source data 4.** Original images for Western blots in *Figure 4F*.

**Source data 5.** Original images for Western blots in *Figure 4I*.

**Source data 6.** Original images for Western blots in *Figure 4K*.

**Figure supplement 1.** Flow cytometry assay demonstrated that insulin receptor substrate 4 (IRS4) knockout had no effect on cell survival, but significantly delayed cell proliferation.

**Figure supplement 2.** PI3K-AKT-mTOR signaling pathway plays a critical role in controlling the proliferation of OVCAR-5 ovarian cancer cells.

---

potential of cell proliferation in *IRS4*-KO ovarian cancer cells (*Figure 4H*). Given the technical challenge in obtaining single clone of *IRS4* KO in OVCAR-3 cells, we took alternative shRNA knockdown strategy (*Figure 4I*). Consistent with KO studies in OVCAR-5 cells, loss of *IRS4* in OVCAR-3 cells also led to growth retardation (*Figure 4J*), highlighting the key role of IRS4 in maintaining ovarian tumor cell growth.

We explored further and compared the difference(s) in downstream signaling pathways upon IRS4 expression alteration. Interestingly, ectopic expression of IRS4 significantly elevated the level of phospho-AKT in HEY and OVCAR-8 cells (*Figure 4B and D*). Consistently, we also observed a dramatic decrease of phospho-AKT in both OVCAR-5 *IRS4*-KO cells (*Figure 4F*) and OVCAR-3 *IRS4*-KD cells (*Figure 4I*), indicating a tight correlation between PI3K-AKT pathway and ovarian cancer cell proliferation. To confirm this, we treated OVCAR-5 WT cells with PI3K inhibitor LY294002 (*Figure 4K–L*) and demonstrated that the inhibition of the PI3K-AKT pathway in OVCAR-5 cells indeed decreased the phosphorylation level of AKT and cell growth in a concentration-dependent manner. These results together strongly implied an indispensable role of IRS4 in connecting the PI3K-AKT signaling pathway and proliferation of ovarian cancer cells.

To further address the importance of PI3K-AKT signaling pathway in regulating ovarian cancer cell growth, we performed a CTG-based compound library screening assay in OVCAR-5 WT ovarian cancer cells. This compound library contains 198 small molecule inhibitors targeting key node genes in cancer cell signaling transduction and metabolism. Thirty out of 198 compounds showed proliferation inhibition greater than 75% (*Figure 4—figure supplement 2A*). Among these, 16 compounds targeted the PI3K-AKT-mTOR signaling pathway (*Figure 4—figure supplement 2B*). This result strongly supports the supposition that the PI3K-AKT-mTOR signaling pathway plays a critical role in controlling the

proliferation of OVCAR-5 ovarian cancer cells, and it is of great interest to understand the molecular details of how IRS4 mediates AKT activation.

Interestingly, whereas there was a dramatic decrease of phospho-AKT signal upon IRS4 deletion, we also observed evident elevation in phospho-ERK level in OVCAR-5 *IRS4*-KO cells (*Figure 4F*), indicating a potential compensatory effect of ERK signaling pathway which may contribute to the survival and proliferative capacities of OVCAR-5 cancer cells after IRS4 deletion. To test this hypothesis, we treated OVCAR-5 WT cells with PI3K inhibitor LY294002 along with ERK pathway inhibitor U0126, followed by CTG assay. The inhibition of both PI3K-AKT pathway and ERK pathway in OVCAR-5 cells almost blocked cell growth completely (*Figure 4M*). Therefore, in the absence of IRS4-mediated PI3K-AKT activation, OVCAR-5 ovarian cancer cells would upregulate ERK activity in a compensating manner, thereby enhancing survival and proliferative capacities.

## PIK3R2 was identified as one of the major downstream signaling components of IRS4 by proximity labeling PUP-IT assay

Encouraged by our previous results we attempted to delineate the molecular mechanism by which IRS4 regulates the AKT signaling pathway, and therefore the likely mechanism by which it regulates the cell proliferation of ovarian cancer. We postulated that FER-mediated IRS4 phosphorylation would facilitate signaling molecule recruitment, which is important for AKT activation. To assess this hypothesis, we applied pupylation-based interaction tagging (PUP-IT) system to identify potential interacting proteins of phosphorylated IRS4 (*Figure 5A*). PUP-IT has been reported as a new ligase-mediated proximity labeling technique, with advantages in detecting weak and transient protein-protein interactions under physiological conditions in living cells (*Liu et al., 2018*). Specifically, we focused on the differences in binding proteins between IRS4-WT and multiple IRS4-YF in OVCAR-5 cells. In combination with previous results, we included five single Y to F mutants (Y779F, Y847F, Y921F, Y656F, and Y828F) into this PUP-IT study. We also included the IRS4-5YF mutant, in which all five abovementioned Tyr residues were replaced by Phe.

The results were summarized and illustrated as volcano map (*Figure 5B–G*), with $\text{Log}_2$ (fold change) as the x-axis and $–\text{Log}_{10}$ (p-value) as the y-axis. We highlighted genes with p-value less than 0.05 ($–\text{Log}_{10}$ (p-value) = 1.30) and preferential IRS4-WT binding over IRS4-YF mutant, with fold change greater than 2.30 ($\text{Log}_2$ (fold change) = –1.20). Among all detected proteins, the number of unique peptides of IRS4 ranked at the top (*Supplementary files 10–12*), indicating the robustness of the assay. Compared to the IRS4-WT control, three YF mutants of IRS4 (Y779F, Y847F, and Y921F) showed reduced binding affinity with EBNA1 binding protein 2 (*Figure 5C and E–F*). SUMO3, SLTM, ATP6V1G1, HIST2H3PS2, CXorf23, and C18orf25 also showed reduced binding affinity with two of the YF mutants of IRS4. Notably, we observed that PIK3R2 (the regulatory subunit of PI3K) preferred to bind with WT rather than the Y779F mutant of IRS4 (*Figure 5C*). Interestingly, this preferential association of PIK3R2 appeared again when comparing interaction proteins between WT and 5YF mutant of IRS4 (*Figure 5G*). The results from these PUP-IT assays strongly suggest that the FER-triggered tyrosine phosphorylation of IRS4 facilitated the recruitment of PIK3R2 and that Tyr779 was the major contributing residue.

To further validate the PUP-IT results, we performed an immunoprecipitation experiment to assess the essentiality of IRS4 phosphorylation on Y779 site in recruiting PIK3R2. Compared to the WT control, the Y779F mutation of IRS4 exhibited dramatically decreased binding affinity with PIK3R2, and subsequently decreased AKT phosphorylation as well (*Figure 5H*), indicating the importance of the tyrosine phosphorylation of the 779 residue (as mediated by FER) in recruiting PIK3R2 and AKT activation. PIK3R2 (also known as p85β) is a key regulatory subunit of PI3K in the PI3K-AKT signaling pathway (*Vallejo-Díaz et al., 2019*). Unlike *PIK3R1* (*p85α*) whose function is tumor suppressive, *PIK3R2* plays a role as oncogene (*Vallejo-Díaz et al., 2019*). Further co-immunoprecipitation assay demonstrated that both PIK3R1 and PIK3R2 can form complex with IRS4, and the presence of PIK3R1 wouldn't block the formation of the IRS4-PIK3R2 complex (*Figure 5—figure supplement 1A*). More interestingly, unlike PIK3R2, PIK3R1 could form complex with both WT and Y779F mutant of IRS4 with equivalent binding affinity (*Figure 5—figure supplement 1B*). Therefore, distinctive binding mechanisms were harnessed by PIK3R1 and PIK3R2, respectively, to form complex with IRS4.

Taken together, both PUP-IT assay and biochemical pull down assay consistently demonstrated that FER-mediated tyrosine phosphorylation of IRS4 at Tyr779 enhanced the recruitment of PIK3R2

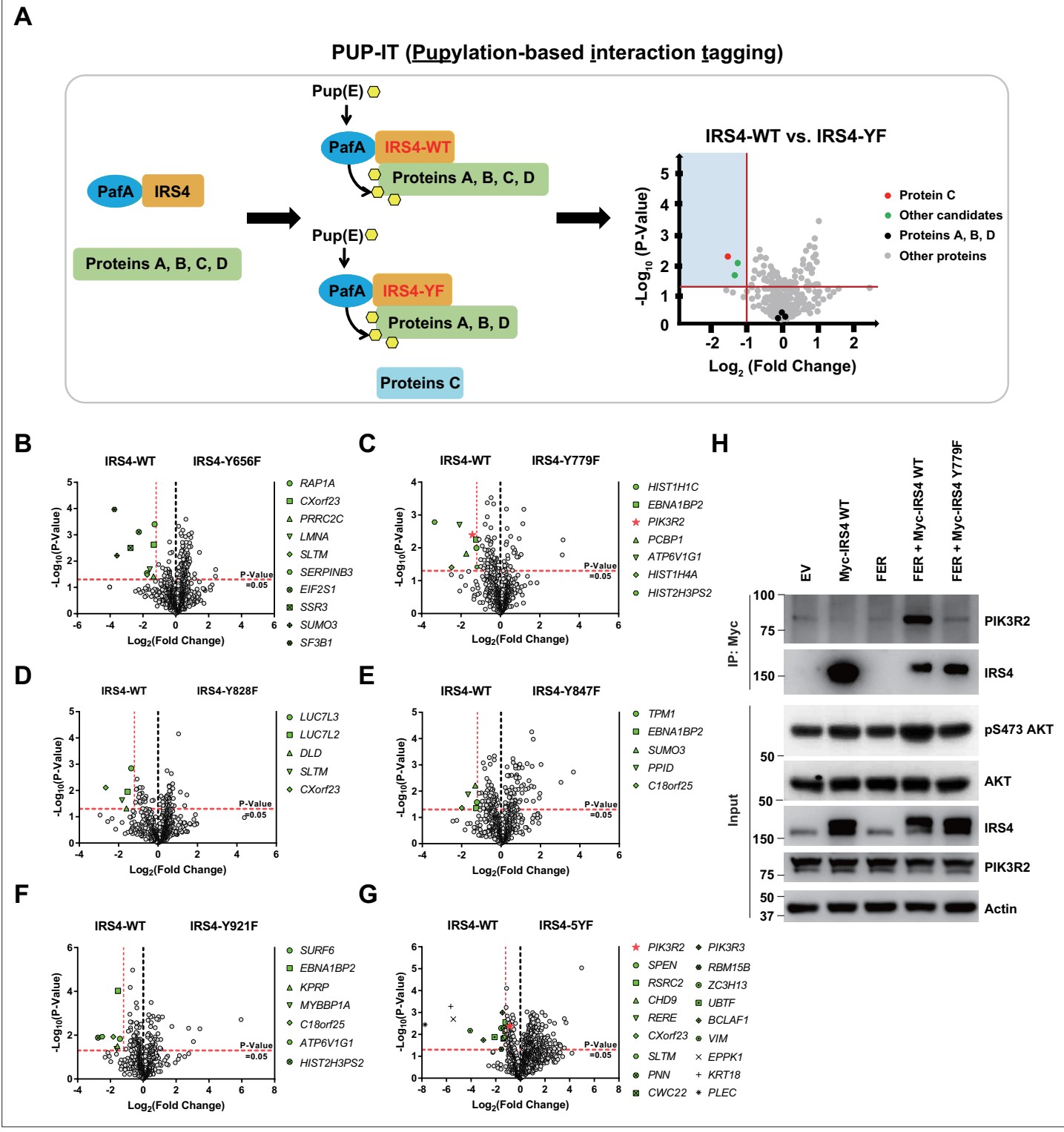

**Figure 5.** PIK3R2 was identified as one of the major downstream signaling components of insulin receptor substrate 4 (IRS4) by proximity labeling pupylation-based interaction tagging (PUP-IT) assay. (**A**) Scheme of potential interacting proteins identification for tyrosine phosphorylated IRS4 using PUP-IT proximity labeling platform. Final results were illustrated with volcano plots drawn by GraphPad Prism. (**B–G**) PUP-IT results from mass spectrometry analysis for each potential interacting protein were showed in volcano plots, with LFQ intensity (three replicates) fold change (x-axis) and p-value change (y-axis). Fold change >2.3 was regarded as difference and p-value < 0.05 calculated by Student's t-test was regarded as statistical significance. Genes with significant binding differences were presented as green or red (*PIK3R2*) dots. (**H**) Myc-tagged IRS4 or its Y779F mutant was

*Figure 5 continued on next page*

*Figure 5 continued*

co-expressed with FER in HEK293FT cells, as indicated, followed by immunoprecipitation with resin against Myc-tag and immunoblotting with IRS4 and PIK3R2 antibodies. The blot was also probed with antibodies against IRS4, PIK3R2, actin, pS473-, and total-AKT in input samples (*Figure 5—source data 1*).

The online version of this article includes the following source data and figure supplement(s) for figure 5:

**Source data 1.** Original images for Western blots in *Figure 5H*.

**Figure supplement 1.** Both PIK3R1 and PIK3R2 formed complex with insulin receptor substrate 4 (IRS4), and the presence of PIK3R1 didn't block the formation of the IRS4-PIK3R2 complex.

**Figure supplement 1—source data 1.** Original images for Western blots in *Figure 5—figure supplement 1A*.

**Figure supplement 1—source data 2.** Original images for Western blots in *Figure 5—figure supplement 1B*.

and activation of the PI3K-AKT signaling pathway, providing new insights into signaling events that underlie cell proliferation in ovarian carcinoma cells.

## FER-mediated PIK3R2 recruitment by IRS4 is crucial to ovarian cancer cell proliferation in vitro and tumorigenesis in vivo

To assess the endogenous binding between FER and IRS4, we performed reciprocal co-immunoprecipitation assay in OVCAR-5 cell line, which has high expression of both proteins (*Figure 4A*; *Fan et al., 2016*). Compared to anti-IgG control, the OVCAR-5 cell lysates with anti-FER or anti-IRS4 antibody showed the robust interaction between FER and IRS4 at endogenous level (*Figure 6A*). Such interaction was disappeared upon CRISPR-Cas9-mediated KO of either *FER* or *IRS4* (*Figure 6B*).

To further pursue the role of the kinase FER in phosphorylating IRS4 in ovarian cancer cells, we generated *FER* KO OVCAR-5 ovarian cancer cells by CRISPR-Cas9 and tested whether the global tyrosine phosphorylation of IRS4 would be affected upon FER loss. We observed decreased tyrosine phosphorylation of IRS4 after tandem IRS4 immunoprecipitation and pTyr immunoblotting (*Figure 6C*). The association of IRS4 and PIK3R2 was also decreased in FER-deficient OVCAR-5 ovarian cancer cells (*Figure 6C*). IGF1R indeed phosphorylated IRS4 in the presence of ligand IGF1 (*Figure 6—figure supplement 1A-B*). However, small molecular IGF1R inhibitor BMS-536924 didn't decrease tyrosine phosphorylation of IRS4 in the absence of IGF1 (*Figure 6—figure supplement 1A-B*), indicating FER-mediated phosphorylation and activation of IRS4 in an IGF1-independent manner. Moreover, IGF1R phosphorylated IRS4 probably in a Tyr779-independent manner, since mutating tyrosine 779 to phenylalanine didn't decrease phosphorylation level of IRS4 mediated by IGF1R receptor tyrosine kinase (*Figure 6—figure supplement 1C*).

In a screen of 586 compounds, TAE684 has been identified as a potent inhibitor against FES, the family member of FER (*Hellwig et al., 2012*). The high similarity between FER and FES inspired us to evaluate if TAE684 exhibits an equivalent inhibitory effect on FER. We first overexpressed FER in HEK293FT cells, followed by TAE684 treatment. We used Tyr402 auto-phosphorylation signal as a readout for measurement of FER kinase inhibition. As shown in *Figure 6—figure supplement 2A*, TAE684 can robustly inhibit the auto-phosphorylation of FER at Tyr402, with $IC_{50}$ around 8.8 nM. Interestingly, compared to lysates from myeloid leukemia cell HL-60, we detected no FES expression in ovarian cancer cell lines used in this study (*Figure 6—figure supplement 2B*). Of note, ovarian cells treated with TAE684 did show dose-dependent inhibition on the kinase activity of FER, as illustrated by pY402 FER blotting analysis (*Figure 6D*). Meanwhile, we witnessed the reduced global tyrosine phosphorylation of IRS4, as well as decreased binding between IRS4 and PIK3R2 in the presence of TAE684 (*Figure 6D*). Of note, the kinase activity of AKT was largely dampened upon TAE684 treatment (*Figure 6D*). Collectively, genetic ablation or pharmacological inhibition of tyrosine kinase FER in ovarian cancer cells leads to decreased global tyrosine phosphorylation and PIK3R2 recruitment of IRS4, which is consistent with our ectopic overexpression studies in HEK293FT cells (*Figure 5H*).

Our previous observations have already demonstrated a retarded cell proliferation in IRS4-deficient ovarian cancer cells (*Figure 4G–H , and J*). To further evaluate the importance of tyrosine phosphorylation of IRS4, we re-expressed these Y to F mutants, along with the WT, in OVCAR-5 *IRS4*-KO ovarian cancer cells. Whereas re-expression of WT IRS4 in *IRS4*-KO cell line fully recovered the growth defect, 5YF mutants of IRS4 failed to rescue cell proliferation (*Figure 6E*). We did observe a certain extent of growth defect rescue with cells re-expressing in the Y779F mutant of IRS4, but these cells still showed

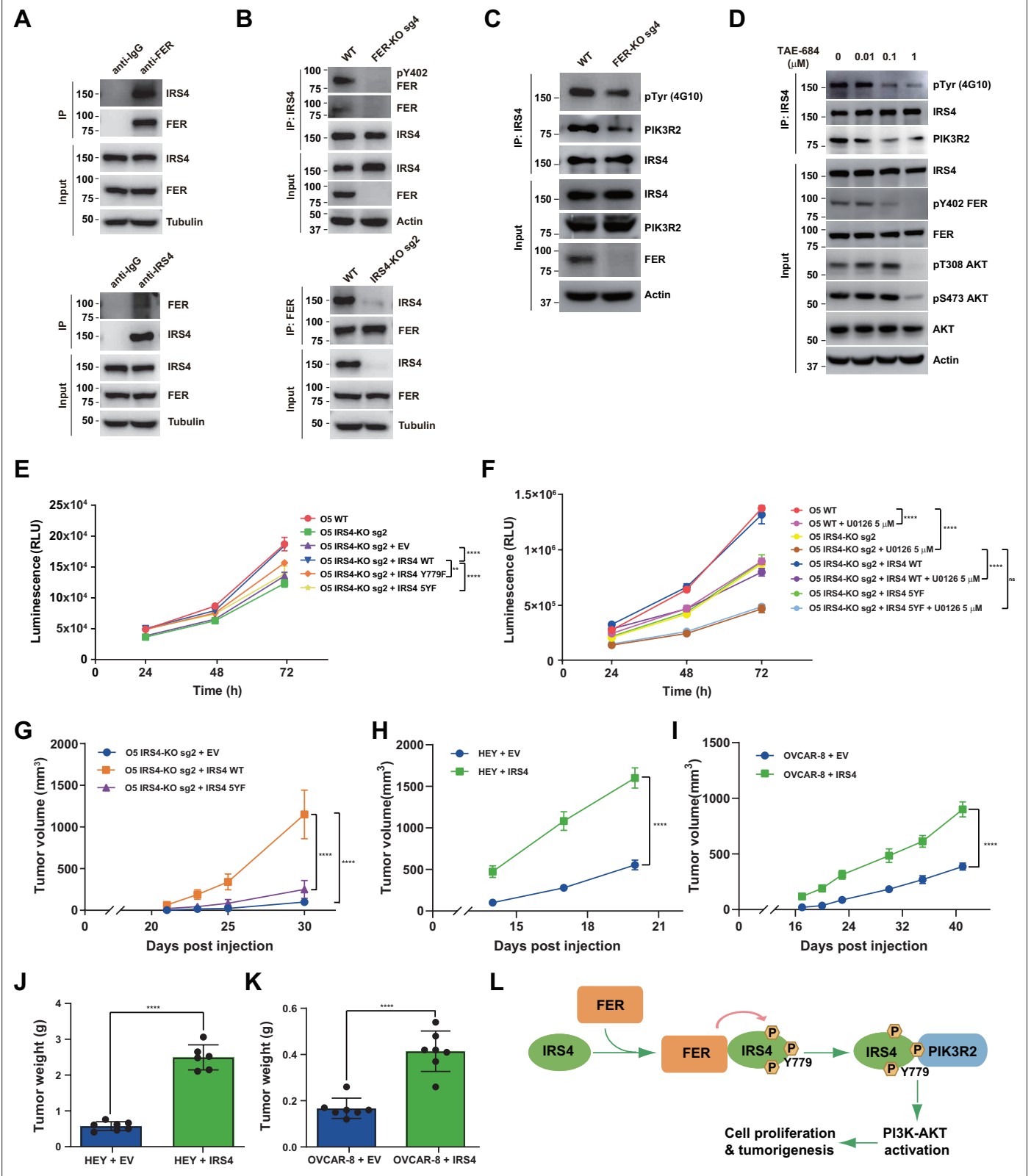

**Figure 6.** FER-mediated PIK3R2 recruitment by insulin receptor substrate 4 (IRS4) is crucial to ovarian cancer cell proliferation in vitro and tumorigenesis in vivo. (**A**) Reciprocal co-immunoprecipitation assay in OVCAR-5 cell line to demonstrate endogenous interaction between FER and IRS4 (*Figure 6—source data 1*). (**B**) Co-immunoprecipitation assay in parental, *FER*-KO, and *IRS4*-KO OVCAR-5 cell lines to demonstrate endogenous interaction between FER and IRS4 (*Figure 6—source data 2*). (**C**) OVCAR-5 wild-type (WT) and OVCAR-5 *FER*-KO cell lysates were harvested and immunoblotted

*Figure 6 continued on next page*

*Figure 6 continued*

for pTyr (4G10), PIK3R2, and IRS4. Actin was probed as loading control. After IRS4 was immunoprecipitated from cell lysates, the global tyrosine phosphorylation of IRS4 and co-immunoprecipitation of PIK3R2 were examined with pTyr (4G10) and PIK3R2 antibodies (*Figure 6—source data 3*). (**D**) OVCAR-5 cells treated with TAE684 (0.01, 0.1, and 1 μM) for 24 hr were lysed, and the expressions of IRS4, pY402-FER, FER, pS473-AKT, pT308-AKT, and actin were detected by immunoblotting as indicated. Endogenous IRS4 immunoprecipitation followed by pTyr and PIK3R2 immunoblotting analysis were also performed (*Figure 6—source data 4*). (**E**) CellTiter-Glo cell viability assay was conducted to evaluate cell proliferation after rescue expression of WT or YF IRS4 in OVCAR-5 *IRS4*-KO cells. OVCAR-5 *IRS4*-KO cells and OVCAR-5 *IRS4*-KO cells expressing empty vector (EV) were included as negative controls. The parental OVCAR-5 cells were used as a positive control. Results represented means ± SEM from three replicates. Significance was determined with two-way ANOVA. \*\*$p = 0.0024$; \*\*\*\*$p < 0.0001$. (**F**) Cell Titer-Glo cell viability assay was conducted to evaluate cell growth capacity after rescue expression of WT or 5YF IRS4 in OVCAR-5 *IRS4*-KO cells exposed to ERK pathway inhibitor U0126 (5 μM) treatment. Results represented means ± SD from five replicates. Significance was determined with two-way ANOVA. ns = no significance; \*\*\*\*$p < 0.0001$. (**G**) After subcutaneous injection of OVCAR-5 *IRS4*-KO cells rescued with EV (n = 5), IRS4-WT (n = 4), and IRS4-5YF (n = 7), respectively, in the xenograft NSG mouse model (NSG mice were randomly assigned and divided into three groups), tumor volumes were measured with calipers at the indicated time intervals. Results represent mean ± SEM. Significance was determined with two-way ANOVA. \*\*\*\*$p < 0.0001$. (H and I) Subcutaneous injections were performed with IRS4-negative cells (HEY or OVCAR-8) with ectopic expression of empty vector (n = 7 for both HEY and OVCAR-8) or IRS4 (n = 6 for HEY, n = 7 for OVCAR-8), respectively. Tumor volumes were measured with calipers at the indicated time intervals. Results represent mean ± SEM. Significance was determined with two-way ANOVA. \*\*\*\*$p < 0.0001$. (J and K) Tumors in (I-J) were dissected at endpoint and weighed. Results represent mean ± SD. Significance was determined with t-test. \*\*\*\*$p < 0.0001$. (**L**) Working model: FER binds directly to IRS4, and phosphorylates its several tyrosine residues. FER-mediated phosphorylation of Tyr779 on IRS4 enhances recruitment of PIK3R2/p85β, the regulatory subunit of PI3K, and promotes PI3K-AKT signaling pathway, which eventually leading to cell proliferation and tumorigenesis in ovarian cancer.

The online version of this article includes the following source data and figure supplement(s) for figure 6:

**Source data 1.** Original images for Western blots in *Figure 6A*.

**Source data 2.** Original images for Western blots in *Figure 6B*.

**Source data 3.** Original images for Western blots in *Figure 6C*.

**Source data 4.** Original images for Western blots in *Figure 6D*.

**Figure supplement 1.** Pharmacological inhibition of IGF1R didn't decrease tyrosine phosphorylation of insulin receptor substrate 4 (IRS4).

**Figure supplement 1—source data 1.** Original images for Western blots in *Figure 6—figure supplement 1A*.

**Figure supplement 1—source data 2.** Original images for Western blots in *Figure 6—figure supplement 1B*.

**Figure supplement 1—source data 3.** Original images for Western blots in *Figure 6—figure supplement 1C*.

**Figure supplement 2.** OVCAR-5 cells treated with TAE684 showed a dose-dependent growth inhibition.

**Figure supplement 2—source data 1.** Original images for Western blots in *Figure 6—figure supplement 2A*.

**Figure supplement 2—source data 2.** Original images for Western blots in *Figure 6—figure supplement 2B*.

significantly delayed proliferation rate compared to the parental ovarian cancer cells (*Figure 6E*). We repeated the whole experiment in the presence of ERK pathway inhibitor U0126. Consistently, *IRS4* KO OVCAR-5 cells treated with U0126 demonstrated profound inhibition in cell growth (*Figure 6F*). Most importantly, re-expression of WT IRS4 in *IRS4*-KO cells exposed to U0126 treatment recovered the growth capacity to the same level as U0126 treatment alone, whereas 5YF mutant of IRS4 failed to rescue cell proliferation (*Figure 6F*). In addition, OVCAR-5 cells treated with TAE684 showed a dose-dependent growth inhibition, as illustrated by CTG assay (*Figure 6—figure supplement 2C-D*). These results demonstrate that the FER kinase-mediated tyrosine phosphorylation of IRS4 plays a key function in controlling cell proliferation in ovarian cancer.

The significant difference between the WT and YF mutants of IRS4 in regulating cell proliferation in vitro prompted us to extend the comparison of ovarian tumorigenesis in vivo. We adopted a xenograft mouse model with subcutaneous injection of OVCAR-5 *IRS4*-KO cells which was rescued with either an empty vector, or a WT or 5YF mutant of IRS4. Compared to mice injected with OVCAR-5 *IRS4*-KO cells rescued with WT IRS4, we observed significantly delayed tumor formation in mice injected with OVCAR-5 *IRS4*-KO cells with an empty vector (*Figure 6G*). Consistent with our previous findings in cell cultures, the tumor growth in mice injected with OVCAR-5 *IRS4*-KO cells rescued with 5YF IRS4 was profoundly delayed compared to WT control (*Figure 6G*), further emphasizing the necessity of the tyrosine phosphorylation of key residues in IRS4 for ovarian tumorigenesis and progression. Consistently, ectopic expression of IRS4 in HEY and OVCAR-8 cells significantly accelerated tumor formations compared to parental controls (*Figure 6H–K*).

## Aberrantly high expression of IRS4 was inversely correlated with prognosis in patients with ovarian cancer

To investigate the expression pattern of IRS4 in human organs and tissues, we first analyzed the RNA-seq data from Human Protein Atlas (HPA) dataset (http://proteinatlas.org). Interestingly, IRS4 shows the highest mRNA transcript abundance in the ovaries, followed by the thyroid gland and endometrium (*Figure 7A*). We further compared the protein expression levels of IRS4 among tissue microarrays of both normal ovaries and malignant ovarian carcinoma in the HPA database. In line with our findings in cell cultures, we observed a higher expression of IRS4 in ovarian cancer patient samples (*Figure 7B*).

In addition, we collected 10 cases of normal ovary samples and 18 cases of malignant ovarian carcinomas samples to further explore the expression differences in IRS4 by immunohistochemistry staining (*Figure 7C*). The antibody used was quite specific for IRS4, since no signal was observed in xenograft tumor sample derived from *IRS4*-KO OVCAR-5 cells rescued with an empty vector (*Figure 7—figure supplement 1*). In accordance with the result from HPA database, the expression levels of IRS4 in ovarian tumor samples were significantly elevated compared to normal control samples (*Figure 7D*).

To further assess the relationship between IRS4 expression and tumor progression, we analyzed clinical data from over 600 ovarian cancer patients (http://www.kmplot.com) and plotted the overall survival curves for both the IRS4-high and the IRS4-low cohorts. The result demonstrated that a lower expression of IRS4 was correlated to longer overall survival in patients with ovarian cancer (*Figure 7E*). In conclusion, IRS4 was significantly overexpressed in ovarian cancers and its upregulation was inversely correlated with survival and prognosis in ovarian cancer patients.

## Discussion

Insulin receptor substrates (IRSs) are cytoplasmic adaptor proteins that participate in the signal transduction process of various receptor tyrosine kinases, including as insulin receptor (IR) and insulin-like growth factor 1 receptor (*Taniguchi et al., 2006*). The IRS family consists of four closely related members IRS1–IRS4 and two distant relatives IRS5/DOK4 and IRS6/DOK5. Although the members of the IRSs family are similar in overall structure and possess high homology, there are differences in tissue and subcellular distribution, and interactions with protein molecules containing the SH2 domain, which enable the six IRS proteins to have different biological characteristics and mediate different signal transduction pathways.

Combining mass spectrometry analysis with biochemical and biological approaches, we have revealed one of the IRS family members, IRS4, as a novel substrate of non-receptor tyrosine kinase FER. FER binds directly to IRS4 (*Figure 2*) and phosphorylates several tyrosine residues on IRS4 (*Figure 3*). The proximity labeling PUP-IT assay further confirmed that FER-mediated phosphorylation of Tyr779 on IRS4 is critical in recruiting PIK3R2/p85β, the key regulatory subunit of PI3K kinase (*Figure 5*). While ectopic overexpression of FER dramatically increased tyrosine phosphorylation and PIK3R2 recruitment of IRS4, CRISPR-Cas9 directed KO or pharmacological inhibition of the endogenous kinase in ovarian cancer cells remarkably reduced tyrosine phosphorylation and PIK3R2 recruitment of IRS4. The current working model is present in *Figure 6L*.

By replacing two SHP2 binding COOH-terminal tyrosines of IRS1 to phenylalanine, Morris White and his colleagues have demonstrated that this mutant form of IRS1 failed to bind SHP2, and exhibited increased tyrosine phosphorylation, phosphatidylinositol 3′-kinase binding, and activation of protein synthesis in response to insulin. These results clearly suggest that SHP2 attenuates the phosphorylation and downstream signal transmission of IRS1 and that the interaction of IRS1 and SHP2 is an important regulatory event which attenuates insulin metabolic responses (*Myers et al., 1998*). By using liver-specific *SHP2*-KO mice, Fawaz Haj and his colleagues were able to show that SHP2 is a negative regulator of hepatic insulin action, and its deletion enhances the activation of PI3K/AKT pathway downstream of the IR (*Matsuo et al., 2010*). Unlike IRS1 and IRS2, IRS4 has no SHP2 binding motif to recruit tyrosine phosphatase SHP2 (*Wauman et al., 2008*; *Fantin et al., 1998*). This unique feature allows IRS4 to maintain constitutive hyperactivation of the PI3K-AKT signaling pathway (*Wauman et al., 2008*; *Fantin et al., 1998*; *Cuevas et al., 2007*), leading to growth factor-independent cell proliferation and tumorigenesis in mammary epithelial cells (*Ikink et al., 2016*). Genetic ablation of IRS4 with either CRISPR-Cas9 or shRNA almost completely abolished the activation of AKT kinase

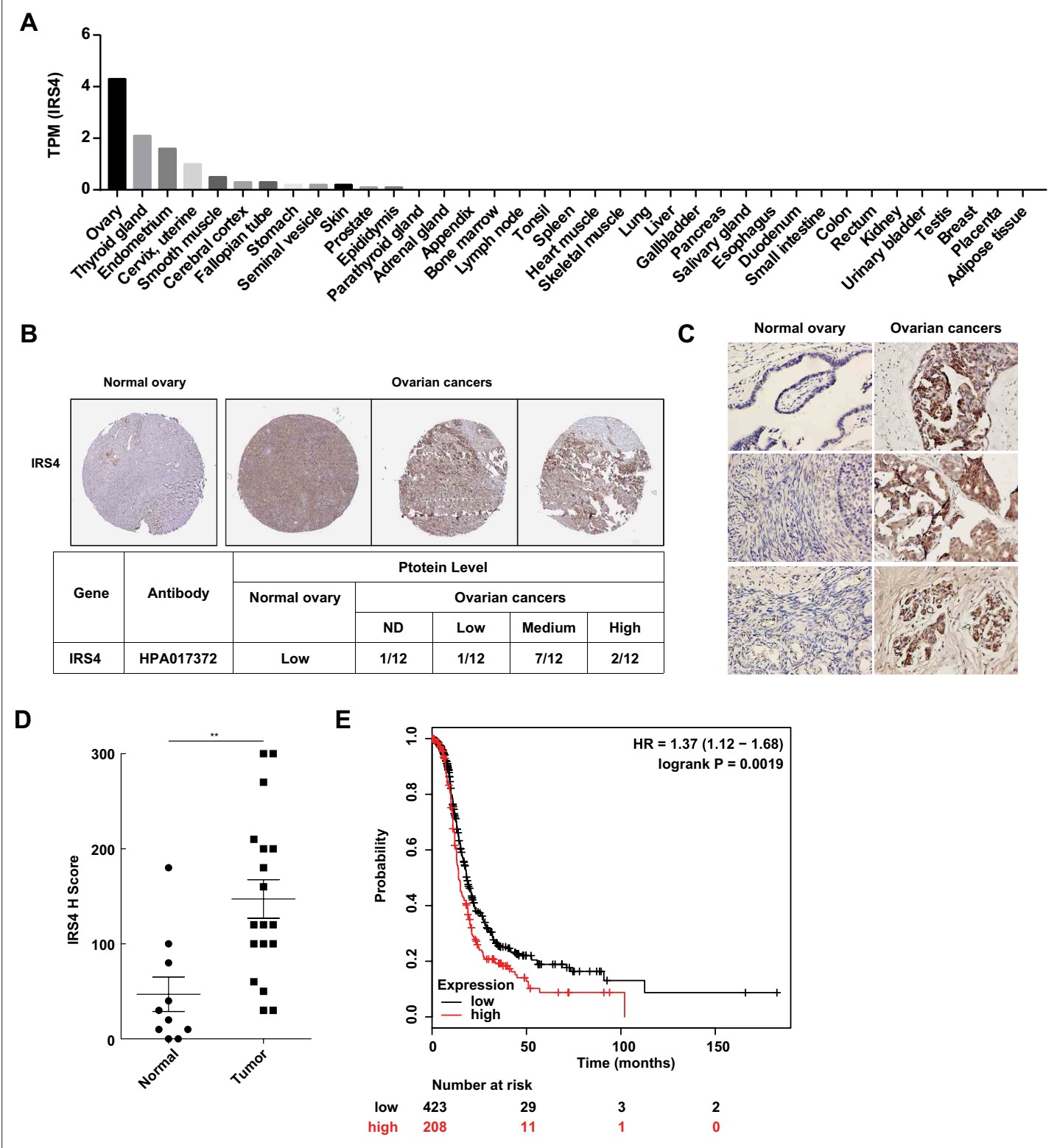

**Figure 7.** Aberrantly high expression of insulin receptor substrate 4 (IRS4) was inversely correlated with prognosis in patients with ovarian cancer. (**A**) mRNA expression profile of IRS4 in multiple human tissues based on RNA-seq tissue data from the Human Protein Atlas (HPA) dataset (http:// proteinatlas.org). Data were reported as mean pTPM (protein-coding transcripts per million), corresponding to mean values of the different individual samples from each tissue. (**B**) The expression of IRS4 in the sections of normal ovary and malignant ovarian carcinoma according to representative tissue microarrays cores from the HPA database. (**C**) Immunohistochemistry staining for IRS4 protein in normal ovaries (n = 10) and malignant ovarian

*Figure 7 continued on next page*

eLife Research article

Biochemistry and Chemical Biology | Cancer Biology

carcinomas (n = 18) samples. Representative images were shown. (**D**) Summary and statistical analysis of immunohistochemistry staining status of IRS4 H score between normal ovaries (n = 10) and malignant ovarian carcinomas (n = 18) samples. Results represented means ± SD. Significance was determined with Student's t-test. **p = 0.0029. (**E**) Overall survival from over 600 ovarian cancer patients among previously published datasets by the km Plotter (http://www.kmplot.com). IRS4 expression was stratified as high versus low against median expression.

The online version of this article includes the following figure supplement(s) for figure 7:

**Figure supplement 1.** Insulin receptor substrate 4 (IRS4) antibody was specific for IRS4.

(*Figure 4F and I*) and dramatically delayed cell proliferation (*Figure 4G–H , and J*). Interestingly, these cells were very sensitive to inhibitors against the PI3K-AKT-mTOR pathway (*Figure 4K–L* and *Figure 4—figure supplement 2*). This suggests that ovarian cells with high IRS4 expression may depend on IRS4-mediated PI3K-AKT activation for proliferation and survival. Any pharmacological perturbation of this pathway would benefit therapeutic outcomes for patients suffered from this deadly disease. Furthermore, our results (*Figures 4H, M and 6F*) also suggested simultaneously targeting both IRS4-mediated PI3K-AKT and ERK pathways may deliver a more effective strategy to treat ovarian cancer.

There are seven Y-X-X-M motifs on IRS4, which have been speculated as potential binding sites for the regulatory subunit of PI3K. Our results from mass spectrometry and site-directed mutagenesis analysis revealed five major Tyr residues as potential substrates for FER kinase: Tyr656, -779, -828, -847, and -921 (*Figure 3C–E*). However, it also should not be ignored that the 1–550aa truncation mutant of IRS4 also possessed weak but detectable phosphorylation signals in the presence of FER kinase. Actually, Tyr487 of human IRS4 also resembles YXXM motif that upon phosphorylation is predicted to bind SH2 domains in the p85 regulatory subunit of PI3K. Among five major Tyr residues, Tyr779, -828, and -921 reside in YXXM motifs to create potential PI3K binding sites. Proximity labeling PUP-IT assays further demonstrated that Tyr779 was the major site responsible for PIK3R2/p85β recruitment on IRS4 (*Figure 5B–F*) and this result was confirmed by co-immunoprecipitation (*Figure 5H*). Our data provides compelling evidence to decipher the molecular details on PIK3R2/p85β association with IRS4.

It has been reported that FER is significantly upregulated in ovarian cancer cell lines and ovarian cancer tumor samples, compared to normal controls and that its downregulation by RNAi results in substantial attenuation of tumor cell migration, invasion, and metastasis, with little change in primary tumor growth (*Fan et al., 2016*). In the current study, tumor growth was significantly suppressed in the absence of IRS4, and re-expressing phosphorylation-defective mutant of IRS4 failed to rescue proliferation rate of the tumor cells. There is one determining factor that should be taken into consideration for aligning these two studies. In the previous report, CAOV4 cells have been used in the majority of in vivo assays, which expresses no IRS4. However, two cell lines used in this study, OVCAR-5 and OVCAR-3, have high IRS4 expression. We think this key difference could explain the different functional output of FER in both experimental context: In the absence of IRS4, FER regulates ovarian cancer cell motility and invasiveness mainly through MET-GAB1-SHP2-ERK1/2 signaling pathway, with MET and GAB1 as its substrates; in IRS4-positive ovarian cancer cells, FER-mediated phosphorylation of Tyr779 enables IRS4 to recruit PIK3R2/p85β, the regulatory subunit of PI3K, and activate the PI3K-AKT pathway for proliferation.

Besides IRS4 and IGF1R (*Stanicka et al., 2018*), our mass spectrometry analysis captured several interesting FER-interacting hits with functions of solute transportation, including *SLC25A5, SLC25A6 SLC3A2, ATP1A1,* and *ATP2A2* (*Figure 1C*). Notably, tyrosine phosphorylation has been reported as an important layer of regulation for transporter proteins' stability (*Loureiro et al., 2019*) and activity (*Xu et al., 2018*; *Zhao et al., 2016*; *Hedges et al., 2013*; *Wertheimer et al., 2008*). We also identified CTLC (Clathrin Heavy Chain) and COPA (COPI Coat Complex Subunit Alpha) as potential FER-associated proteins, the functions of which are involved in both Clathrin-dependent and -independent intracellular trafficking (*Figure 1C*). In addition, regulators in cytosol-nucleus shuttle, for example, *XPO1* and *IPO4*, were also ranked high in the hit list (*Figure 1C*). Considering the well-documented function of FER kinase in vesicle trafficking and cell motility, these findings will definitely shed new light on molecular mechanism on FER's function.

In summation, our study has demonstrated IRS4 as a novel substrate for non-receptor tyrosine kinase FER. This kinase-substrate regulatory mode between FER and IRS4, which leads to PIK3R2

recruitment and AKT activation, is critical for ovarian tumor cell growth. This work expounds on the versatile functions of the FER kinase, especially within ovarian cancer, and highlights the unmet need to develop a small molecule inhibitor of the kinase to benefit patients.

## Materials and methods
### Cell culture and chemical reagents
OVCAR-5 and OVCAR-3 cell lines were obtained from Robert Lucito lab in Cold Spring Harbor Laboratory. OVCAR-8 cell line was obtained from Ahmed Ashour Ahmed lab in Ovarian Cancer Cell Laboratory, University of Oxford. HEY cell line was obtained from Robert C Bast lab in MD Anderson Cancer Centre. Human embryonic kidney cells 293 (HEK293T) were obtained from the cell bank of CAS (Shanghai). The cell lines have been tested to be free of mycoplasma contamination by stand PCR methods. Cells were cultured in Dulbecco's modified Eagle's medium (DMEM, Cellgro) supplemented with 10% fetal bovine serum (Cellgro), 100 units/ml penicillin, and 100 μg/ml streptomycin, and maintained at 37°C in 5% $CO_2$.

PI3K inhibitor LY294002 (S1105), ERK pathway inhibitor U0126 (S1102), Anti-cancer Metabolism Compound Library (L5700), TAE684 (S1108), and IGF1R inhibitor BMS-536924 (S1012) were purchased from Selleck.

### Plasmids
Mammalian expression plasmids used in this study were as follows: pMSCV-FER (a gift from Prof. Peter A Greer, Queen's University, Canada), pEGFP-FER (a gift from Prof. Toshiki Itoh, Kobe University, Japan), FPC1-Myc-IRS4 (IRS4 full-length and IRS4 truncation mutants 1–400, 1–550, 1–699, 1–800, 1–1093, and 200–1257 were gifts from Prof. Kensaku Mizuno, Tohoku University, Japan), pEYFPC1-IRS4 (IRS4 truncation mutants 1–334, 335–400, 200–400, and 401–1257 were gifts from Prof. Kensaku Mizuno, Tohoku University, Japan), pUSE-SRC, pcDNA6-HCK, pHAGE-FYN, pcDNA3.1-LCK, pLPC-BRK, pWZL-BTK, pLV-FER, pcDNA3-IGF1R, PX330-IRS4-sgRNA-Cas9-GFP, PX330-FER-sgRNA-Cas9-GFP, pLKO.1 (Addgene, cat. no. 10878), pGEX-3X-FER (FER truncation mutants 1–446 and 447–822), pCDH-IRS1-FLAG, pCDH-PIK3R1-FLAG, pCMV3-PIK3R2, and PUP-IT-related plasmids (pTet3G-Bio-PupE-IRES-BFP and IRS4-PafA-IRES-puro-GFP). By using pEGFP-FER plasmid as template, we further constructed FER truncation mutants 1–446, 447–822, 447–550, and 563–822, as well as single or multiple mutants of E676R, D684R, and E740R in the kinase domain of FER. By using pLV-FER plasmid as template, we constructed FER QPVY motif (631-634aa) deletion mutants and Y634F mutant. By using FPC1-Myc-IRS4 as template, we further constructed IRS4 PH or PTB domain deletion mutants, IRS4 truncation mutants 1–200, 200–400, and 1–400, as well as all Y to F mutants. Primers used in the construction of related FER and IRS4 mutants were uploaded in *Supplementary file 13*.

### Cell transfection and infection
We followed manufacture protocol of Mirus (TransIT-2020, Mirus Bio) to perform transient transfection. Briefly, cells were plated in a six-well plate 24 hr prior to transfection. When cells reached ~75% confluence, we prepared Mirus:plasmid complexes in Opti-MEM I Reduced Serum Medium (Gibco) and added them into each well. Twenty-four hours later, cells were harvested and lysed for immunoblotting or immunoprecipitation assays.

Cell line with gene stable expression was established by lentiviral infection, followed by GFP sorting or puromycin selection. In brief, lentivirus was generated in HEK293FT cells by co-transfecting gene-containing plasmids, deltaR8.2, and VSVG at a ratio of 3:2:1; 48–72 hr later, supernatants were collected and passed through 0.45 μm filters to remove cell debris. Cleared virus was then added to cells to be infected in the presence of polybrene. Infected cells were either sorted by GFP or selected by puromycin. The effectiveness of infection was confirmed by flow cytometry or immunoblotting with according antibody.

### Protein expression and purification
GST-tagged FER (1-446) and GST-tagged FER (447-822) were expressed in *E. coli* BL21 (DE3). The cells were cultured at 37°C until the OD reached 0.6–0.8 and were induced with 0.3 mM isopropyl β-D-thiogalactoside in LB medium at 16°C overnight. Bacteria were lysed in lysis buffer (50 mM Tris-HCl,

pH 7.5, 250 mM NaCl, 1 mM DTT, and 1× complete protease inhibitor) by high-pressure homogenizer. After centrifugation, the supernatant was incubated with GST beads at 4°C for 2 hr. After washing, GST-tagged proteins were eluted with 10 mM reduced glutathione. Protein concentration was measured using the Bradford assay. Protein purity was assessed by SDS-PAGE and Coomassie blue staining.

## Immunoblotting and immunoprecipitation assay

Cells were lysed in lysis buffer (20 mM HEPES pH 7.5, 150 mM NaCl, 1% Nonidet P-40, 1 mM sodium orthovanadate and 1× complete protease inhibitor cocktail from Roche) at 4°C for 15 min. Total protein concentration was determined by Bradford assay.

For immunoblotting, cellular proteins were harvested, separated by SDS-PAGE, and transferred onto nitrocellulose membranes. Membranes were blocked in 2.5% BSA in TBST (TBS/Tween 20: 20 mM Tris-HCl, pH 7.5, 50 mM NaCl, and 0.1% Tween 20) for 1 hr at room temperature on a shaker and incubated with primary antibody at 4°C overnight. Proteins were detected with horseradish peroxidase-conjugated secondary antibodies (Jackson Laboratory) and ECL (Pierce).

For immunoprecipitation, precleared cell extracts were incubated with the indicated antibody for 4 hr at 4°C with rotation followed by 1 hr of pull-down by 1:1 protein A/G agarose beads. Immunoprecipitates were washed with lysis buffer three times before electrophoresis.

The primary antibodies used in this study were as follows: 4G10 (Millipore); Myc (9E10); FER, GRB2, IRS1, pY1000, Vinculin, FES, phospho- and total-ERK1/2, phospho-Ser473, phospho-Thr308 and total-AKT, phosphor-Tyr1234,1235 and total MET, phosphor-Tyr1131 and total-IGF1R (Cell Signaling Technology); phospho-p38 (Promega); total p38, pTyr1000 and GST (Santa Cruz Biotechnology); pY402 FER (Abcam); IRS4, Actin, Tubulin and FLAG (Sigma); GAPDH (Novus Biologicals); PIK3R2 (Invitrogen); GFP (Abmart). The beads used in this assay were as follows: Streptavidin Magnetic Beads (NEB), EZview(TM) Red anti-c-Myc affinity gel (Sigma), protein A sepharose and protein G sepharose (GE).

## IVK assay

Human IRS4 protein was expressed and purified from HEK293FT cells with Myc-beads. The IVK assay was carried out in assay buffer (25 mM Tris-HCl, pH 7.5, 150 mM NaCl, 10 mM MgCl$_2$) with purified GST-tagged human FER kinase (C-terminal tyrosine kinase domain 541–822aa, purchased from Thermo Fisher Scientific #1871897F). The reaction was initiated by the addition of 10 mM ATP and then carried out at room temperature for 1 hr. 5× SDS loading buffer was added to terminate the reaction. The samples were then analyzed by immunoblotting analysis with anti-IRS4, anti-GST, and anti-pTyr antibodies.

## Cell proliferation assay

CTG luminescent cell viability assay (Promega) was used to evaluate the role of IRS4 in ovarian cancer cell proliferation. In brief, $1.5 \times 10^3$ OVCAR-5 cells, $2 \times 10^3$ OVCAR-3 cells, $1.5 \times 10^3$ HEY cells, or $2 \times 10^3$ OVCAR-8 cells per well were seeded in a 96-well plate, respectively, and grown for indicated time intervals. CTG reagent was added to each well and mixed for ~15 min on an orbital shaker to induce cell lysis followed by luminescence reading.

## Annexin V-FITC and PI double staining assay

Cell apoptosis quantification was performed by Annexin V-FITC/PI Apoptosis Detection kit (C1062, Beyotime). In brief, OVCAR-5 parental or *IRS4*-KO cells were plated into six-well plates. When cells reached 80–90% confluence, they were harvested and washed with PBS. After the addition of 195 μl binding buffer, 5 μl FITC-labeled Annexin V and 10 μl PI were added and incubated for 10–20 min in the dark at room temperature. Cell apoptosis was immediately measured by flow cytometry analysis (LSRFortessa, Becton Dickinson).

## Cell cycle analysis

Cell cycle analysis was performed by cell cycle and apoptosis analysis kit (C1052, Beyotime). Briefly, OVCAR-5 parental or *IRS4*-KO cells were seeded in six-well plates. When cells reached 80–90% confluence, they were harvested and washed with PBS. Cells were fixed in 70% ethanol at 4°C overnight and

washed with PBS again. After the addition of 500 µl buffer supplemented with 10 µl RNase A (50×), cells were stained with 25 µl PI (20×) for 30 min at 37°C. Cell cycle was measured by flow cytometry analysis (LSRFortessa, Becton Dickinson).

## CRISPR-Cas9 system for gene KO

To generate *IRS4* and *FER* KO ovarian cancer cell lines using CRISPR-Cas9 system, the CRISPR sgRNA database (http://crispr.mit.edu/) was applied to generate sgRNAs for each gene. The selected sgRNAs were then subcloned into PX330-Cas9-GFP plasmid, followed by transient transfection into ovarian cancer cell lines and FACS for GFP-positive single clone. The KO effect was confirmed by Western blotting analysis against relevant antibodies.

The sgRNAs used were:

> *IRS4* sgRNA#1: 5'-CCATCGCGAAGTATTCGTCT-3',
> *IRS4* sgRNA#2: 5'-TATAGGGTGATCACGCGCCG-3',
> *FER* sgRNA#4: 5'-AGAGTTTGATACTTCCTTAC-3'.

## shRNA knockdown

The plasmid pLKO.1 (Addgene, cat. no. 10878) was used as shRNA construct backbone. Lentiviral transduction-based shRNA delivery was performed as previously described (*Fan et al., 2016*). In brief, HEK293FT cells were used for virus packaging. DeltaR8.2, VSVG, and shRNA plasmids were co-transfected into HEK293FT cells with Mirus transfection reagents. Cell culture suspension which contained virus was collected 48 hr after transfection. The OVCAR-5 and OVCAR-3 cells were infected and selected with 2 µg/ml puromycin for 2 days.

The shRNAs used were:

> *IRS4* shRNA: 5'- CCGGGCTGGTTTCAACCTGTTGCTACTCGAGTAGCAACAGGTTGAAAC CAGCTTTTTG -3'

## PUP-IT assay

The experimental procedure was modified based on previous study (*Liu et al., 2018*). To generate inducible Pup (iPup) cell lines, we first produced lentivirus with Pup (E) plasmid Bio-Pup (E)-IRES-BFP within the Tet-On 3G inducible expression system (Clontech 631168), and infected OVCAR-5 cells for 48 hr. Subsequently, doxycycline (final concentration 2 µg/ml Selleck, S4136) was added into the culture medium for another 24 hr. BFP-positive cells were then sorted into 96-well plates by flow cytometry for single clone selection. It takes ~3 weeks for cell re-population. After adding doxycycline (final concentration 2 µg/ml) and biotin (final concentration 4 µM) for 24 hr, the BFP expression of each clone was confirmed by flow cytometry, and the expression and modification of Bio-Pup (E) in BFP-positive cells was also confirmed by Western blotting.

To further stably express IRS4 (WT)-PafA or IRS4 (YF)-PafA in iPup OVCAR-5 cells, we subcloned IRS4-WT or IRS4-YF into the PafA-IRES-puro-EGFP plasmid, respectively, and produced lentivirus to infect iPup OVACR5 cells for 48 hr. Cells were placed under puromycin selection (final concentration 2 µg/ml) for generating iPup OVCAR-5 cell lines which stably express IRS4 (WT)-PafA or IRS4 (YF)-PafA, respectively.

IRS4 (WT)-PafA or IRS4 (YF)-PafA expressed iPup OVCAR-5 cells were then grown in 10 cm dishes. We added doxycycline (final concentration 2 µg/ml) and biotin (final concentration 4 µM) to the medium in advance, and induce expression in cells for 24 hr. Then, we harvested cells, and followed the protocol in *Liu et al., 2018*, to prepare sample for mass spectrometry analysis.

Particularly in this study, we compared the binding protein differences between IRS4 (WT) and IRS4 (YF) in OVCAR-5 cells. To obtain reliable and quantitative measurement, each group of samples was triplicated. To analyze the different proteins bound to IRS4 (WT) or IRS4 (YF) in OVCAR-5 cells, we calculated the fold change of LFQ intensity and used the t-test to calculate the p-value. Fold change >2.3 and p-value < 0.05 would be regarded as differences and statistical significance. We used GraphPad Prism to draw the relevant volcano maps.

## Sample preparation, digestion, and mass spectrometry

For identification of phosphorylated tyrosine residues and interacting proteins by mass spectrometry, immunoprecipitates were prepared first as described above. Samples were subjected to SDS-PAGE gel, followed by in-gel trypsin digestion. In brief, gel bands were excised, washed, and dehydrated with 100% acetonitrile. Proteins inside the gel were reduced, alkylated, and finally digested with trypsin overnight at 37°C. The mixture of peptide fragments was extracted with 50% acetonitrile and 1% trifluoroacetic acid followed by 100% acetonitrile. Peptides were vacuum-dried and re-suspended for following mass spectrometry characterization. When samples were subjected to on-beads digestion, the beads in immunoprecipitation were digested with trypsin overnight at 37°C. After cleaning, peptides were vacuum-dried and re-suspended for following mass spectrometry characterization.

Mass spectrometry analysis was performed at the Proteomics Facility in Shanghaitech University. An Easy-nLC 1000 system coupled to a Q Exactive HF (both from Thermo Scientific) was used to separate and analyze peptides. The raw data were processed and searched with MaxQuant 1.5.4.1 with MS tolerance of 4.5 ppm, and MS/MS tolerance of 20 ppm. The UniProt human protein database (release 2016_07, 70,630 sequences) and database for proteomics contaminants from MaxQuant were used for database search.

## Animal work

All study protocols involving mice were approved by the Institutional Animal Care and Use Committee of the ShanghaiTech University and conducted in accordance with governmental regulations of China for the care and use of animals. In the first subcutaneous injection model, $1 \times 10^6$ OVCAR-5 *IRS4*-KO cells with ectopic expression of empty vector (n = 5), IRS4-WT (n = 4), or IRS4-5YF (n = 7), respectively, were suspended in 100 µl of 1:1 mixture with DMEM and growth factor-reduced Matrigel (BD Biosciences) and subcutaneously injected into NSG mice. In the second and third subcutaneous injection models, $1 \times 10^6$ IRS4-negative cells (HEY or OVCAR-8) with ectopic expression of empty vector (n = 7) or IRS4 (n = 6 for HEY, n = 7 for OVCAR-8), respectively, were suspended in 100 µl of 1:1 mixture with DMEM and growth factor-reduced Matrigel (BD Biosciences) and subcutaneously injected into NSG mice. Subcutaneous tumor growth was monitored periodically by measuring tumor volume (in cubic millimeters, formula: volume=width$^2$ × length/2) with calipers.

## Immunohistochemistry staining

All human ovarian normal and tumor tissues were obtained from the Nanjing Maternity and Child Health Care Hospital in compliance with guidelines for informed consent approved by the Hospital's Internal Review Board committee (NFKSL-063). Paraffin-embedded tissues were sectioned and stained with H&E or specific immunohistochemical stains. Stained slides were digitally scanned using the Aperio ScanScope software. H score was used for statistical analysis and calculated as positive staining percentage multiplied by staining strength (*Ren et al., 2014*). Both positive and negative control slides were included. The IRS4 primary antibody used in this assay was from Sigma. The MET primary antibody used in this assay was from Cell Signaling Technology.

## Statistics

The GraphPad Prism (v7.00) was used to perform all statistical analyses including standard Student's t-test or two-way ANOVA multiple comparisons. The sample-size estimation, number of replicates, data presentation, and type of statistical analyses were indicated for each experiment within figure legends. Data were shown as means ± SD or SEM. The following indications of significance were used throughout the manuscript and indicated for each experiment in the figure legends: ns = no significance, **$p < 0.01$, ***$p < 0.001$, ****$p < 0.0001$.

## Acknowledgements

We thank staff members of Mass Spectrometry team at ShanghaiTech University. We also thank staff members of Animal Facility at the National Facility for Protein Science in Shanghai (NFPS), Zhangjiang Lab for providing technical support and assistance in animal work. Funding: This work was supported by the Ministry of Science and Technology of China (2018YFC1004603 to GF), the National Natural Science Foundation of China (31872831 and 32070776 to GF), Science and Technology Commission

of Shanghai Municipality (19JC1413800 to GF), the Shanghai Pujiang program (18PJ1407900 to GF), the Shanghai Shuguang Program (19SG55 to GF) and ShanghaiTech University Startup grant (to GF).

## Additional information

### Funding

| Funder | Grant reference number | Author |
|---|---|---|
| Ministry of Science and Technology of the People's Republic of China | 2018YFC1004603 | Gaofeng Fan |
| National Natural Science Foundation of China | 31872831 | Gaofeng Fan |
| National Natural Science Foundation of China | 32070776 | Gaofeng Fan |
| Science and Technology Commission of Shanghai Municipality | 19JC1413800 | Gaofeng Fan |
| Shanghai Pujiang program | 18PJ1407900 | Gaofeng Fan |
| Shanghai Shuguang Program | 19SG55 | Gaofeng Fan |
| ShanghaiTech University | | Gaofeng Fan |

The funders had no role in study design, data collection and interpretation, or the decision to submit the work for publication.

### Author contributions

Yanchun Zhang, Conceptualization, Investigation, Validation, Visualization, Writing - original draft; Xuexue Xiong, Jiali Zhang, Shengmiao Chen, Yuetong Wang, Li Chen, Linjun Hou, Xi Zhao, Jian Chen, Investigation, Validation; Qi Zhu, Data curation, Visualization; Jian Cao, Dake Li, Investigation, Resources; Piliang Hao, Min Zhuang, Data curation, Methodology; Gaofeng Fan, Conceptualization, Funding acquisition, Investigation, Project administration, Resources, Supervision, Writing - review and editing

### Author ORCIDs

Yanchun Zhang 
Yuetong Wang 
Gaofeng Fan 

### Ethics

All human ovarian normal and tumor tissues were obtained from the Nanjing Maternity and Child Health Care Hospital in compliance with guidelines for informed consent approved by the hospital's Internal Review Board committee (NFKSL-063).

All study protocols involving mice were approved by the Institutional Animal Care and Use Committee of the ShanghaiTech University (P2021-0243) and conducted in accordance with governmental regulations of China for the care and use of animals.

### Decision letter and Author response

Decision letter https://doi.org/10.7554/eLife.76183.sa1
Author response https://doi.org/10.7554/eLife.76183.sa2

## Additional files

### Supplementary files

• Supplementary file 1. Identification of potential FER interactome by in-gel tryptic digestion. Identification of novel substrates for tyrosine kinase FER by pTyr immunoprecipitation and in-gel

tryptic digestion followed by mass spectrometry analysis in *Figure 1B*. The details of raw data analysis and database search have been described in the section of Materials and methods. The analyzed MS data was provided in *Supplementary files 1-6*. Potential FER interactome between 25- and 37 kDa is shown in this file.

• Supplementary file 2. Potential FER Interactome with molecular weight between 37 and 50kDa. Potential FER interactome between 37– and 50 kDa is shown in this file.

• Supplementary file 3. Potential FER Interactome with molecular weight between 50 and 75kDa. Potential FER interactome between 50– and 75 kDa is shown in this file.

• Supplementary file 4. Potential FER Interactome with molecular weight between 75 and 100kDa. Potential FER interactome between 75– and 100 kDa is shown in this file.

• Supplementary file 5. Potential FER Interactome with molecular weight between 100 and 150kDa. Potential FER interactome between 100– and 150 kDa is shown in this file.

• Supplementary file 6. Potential FER Interactome with molecular weight greater than 150kDa. Potential FER interactome above 150 kDa is shown in this file.

• Supplementary file 7. Identification of potential FER interactome by beads digestion. Identification of novel substrates for tyrosine kinase FER by pTyr immunoprecipitation and on beads tryptic digestion followed by mass spectrometry analysis in *Figure 1B*. The details of raw data analysis and database search have been described in the section of Materials and methods. The analyzed MS data is provided in this file.

• Supplementary file 8. Identification of tyrosine phosphorylation sites on insulin receptor substrate 4 (IRS4) by mass spectrometry analysis. Identification of tyrosine residue(s) that undergo phosphorylation in the presence of FER by mass spectrometry analysis in duplicate in *Figure 3D*. The details of raw data analysis and database search have been described in the section of Materials and methods. The analyzed MS data was provided in *Supplementary files 8-9*. First identification of tyrosine phosphorylation sites on IRS4 is shown in this file.

• Supplementary file 9. Identification of tyrosine phosphorylation sites on insulin receptor substrate 4 (IRS4) by mass spectrometry analysis (2nd attempt). Second identification of tyrosine phosphorylation sites on IRS4 is shown in this file.

• Supplementary file 10. Identification of potential interacting proteins of insulin receptor substrate 4 (IRS4) wild-type (WT) and YF mutants by pupylation-based interaction tagging (PUP-IT) system. Exploration for differences in binding proteins between IRS4-WT and multiple IRS4-YF in OVCAR-5 cells in *Figure 5A–G*. The details of raw data analysis and database search have been described in the section of Materials and methods. The analyzed MS data was provided in *Supplementary files 10-12*. The IRS4-WT, IRS4-Y779F, IRS4-Y847F, and IRS4-Y921F interactome is shown in this file.

• Supplementary file 11. Identification of potential interacting proteins of insulin receptor substrate 4 (IRS4) wild-type (WT) and YF mutants (Y656F and Y828F) by pupylation-based interaction tagging (PUP-IT) system. The IRS4-WT, IRS4-Y656F, and IRS4-Y828F interactome is shown in this file.

• Supplementary file 12. Identification of potential interacting proteins of insulin receptor substrate 4 (IRS4) wild-type (WT) and 5YF mutant by pupylation-based interaction tagging (PUP-IT) system. The IRS4-WT and IRS4-5YF interactome is shown in this file.

• Supplementary file 13. Primers used in the construction of related FER and insulin receptor substrate 4 (IRS4) mutants. Primers used in the construction of related FER and IRS4 mutants.

• Transparent reporting form

### Data availability

The raw spectrometry proteomics data have been deposited to the ProteomeXchange Consortium with the dataset identifier PXD022084 (The URL in ProteomeXchange Consortium: http://proteomecentral.proteomexchange.org/cgi/GetDataset?ID=PXD022084. The URL in iProX: https://www.iprox.org/page/project.html?id=IPX0002540000). Analyzed MS data was provided in Supplementary File 1-12. The original files of the full raw unedited gels or blots as well as figures with the uncropped gels or blots with the relevant bands clearly labelled were provided in Original Files of the Gels and Blots as Source data files.Primers were uploaded in Supplementary File 13.

The following dataset was generated:

| Author(s) | Year | Dataset title | Dataset URL | Database and Identifier |
|---|---|---|---|---|
| Zhang Y | 2020 | FER-mediated phosphorylation and PIK3R2 recruitment on IRS4 promotes AKT activation and tumorigenesis in ovarian cancer | http://proteomecentral.proteomexchange.org/cgi/GetDataset?ID=PXD022084 | ProteomeXchange, PXD022084 |

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
