## [Editor Report]

This study was designed to examine the role of the FER/IRS4 pathway in ovarian cancer cells. The authors show that FER causes tyrosine phosphorylation of IRS4 and recruitment of PIK3R2 that subsequently causes activation of AKT. The data presented suggest that pharmacological targeting of FER/IRS4 may be beneficial for the treatment of ovarian cancer.

---

## [Decision Letter]

**Decision letter after peer review:**

[Editors’ note: the authors submitted for reconsideration following the decision after peer review. What follows is the decision letter after the first round of review.]

Thank you for submitting your work entitled "FER-mediated phosphorylation and PIK3R2 recruitment on IRS4 promotes AKT activation and tumorigenesis in ovarian cancer" for consideration by *eLife*. Your article has been reviewed by 3 peer reviewers, one of whom is a member of our Board of Reviewing Editors, and the evaluation has been overseen by a Senior Editor. The reviewers have opted to remain anonymous.

Our decision has been reached after consultation between the reviewers. Based on these discussions and the individual reviews below, we regret to inform you that your work will not be considered further for publication in *eLife*.

This is an interesting study that identified IRS4 as a mediator of FER signaling. However, a number of problems with thus study were identified by the reviewers. First, the study exhibits a reliance on the use of over-expression (rather than endogenous) protein analysis; second, the conclusions presented conflict with prior published work by the senior author (MET etc) and these conflicts are not explained; and third, the relevance of the conclusions to human cancer are not fully established. The specific issues noted by the reviewers are appended to this letter. It is clear that substantial revision of your manuscript will be required. It is the policy of *eLife* to return the manuscript to the authors under these circumstances.

*Reviewer #1:*

This is an interesting paper that demonstrates a role for IRS4 as a target of Fer signaling that leads to PIK3R2 recruitment and Akt activation that may be relevant to ovarian cancer. This pathway is demonstrated to promote the growth of human ovarian cancer cells in in vitro and in a xenograft assay.

The major novelty of this study is the demonstration that this pathway is independent of Met/Gab1, which was previously implicated in Fer-stimulated AKT activation. This is a surprising finding. Otherwise, the study appears to be a relatively standard biochemical analysis with minimal attention to cancer relevance.

There are some problems with the study presented that need to be addressed.

a) All the biochemical studies appear to rely on over-expressed proteins. The study would be improved by showing that the reported protein interactions are detected in assays using endogenous proteins.

b) What is the evidence that IRS4 is a Fer substrate? The data presented seems to be limited to a correlation between Fer over-expression and knockout with IRS4 tyrosine phosphorylation in cultured cells. Is IRS4 phosphorylated by Fer in vitro?

c) Some of the details of experimental procedures are unclear. For example, three different guide RNAs are described for Fer KO cells, yet only a single line of Fer KO cells appears to be presented. Which guide RNA was used? Similarly, three different guide RNAs are described for IRS4 KO lines and cell two lines were reported. Which guide RNAs were used?

d) The tumors in the xenograft study appear to be large (>2,000 m3). The authors should confirm that tumors this large are acceptable to the institutional animal care and use committee.

*Reviewer #2:*

Summary:

This study examines the mechanism by which the FER non-receptor tyrosine kinase contributes to ovarian carcinoma cell growth. Mass spectrometry was used to identify targets of the FER kinase and IRS4 was identified as one of the top candidates. As reported by the authors, high IRS4 expression correlates with worse outcomes in ovarian cancer patients, which emphasizes the significance of this study. Overall the data presented are clear and support a role for IRS4 in growth regulation. However, the data are limited to primarily one cell line and over-expression studies which diminishes the impact. Moreover, the novel mechanistic information gained regarding IRS4 function is limited. For these reasons, publication in *eLife* is considered premature at this time.

1) IRS4 belongs to the family of insulin receptor substrate proteins that have been best characterized for mediating signaling downstream of the insulin and insulin-like growth factor-1 (IGF-1) receptors to regulate organism metabolism and growth, respectively. IGF-1R was also identified as a target of FER in the mass-spec analysis. The authors do not address if the function of IRS4 is dependent upon this FER-mediated phosphorylation of IGF-1R. In addition, the IRS4 tyrosine mutations could also interfere with IGF-1R signaling to impact tumor growth, and this should be investigated to be able to attribute the IRS4-dependent effects to FER.

2) In a previous study (G. Fan et al., Genes and Dev, 2016), ovarian carcinoma cells with or without expression of FER were injected subcutaneously in mice and differences in tumor growth were not observed. However, metastasis to the lungs was decreased. In the current study, tumor growth was suppressed when IRS4 expression was knocked out. This raises concern that the growth-promoting effects of IRS4 are independent of FER. This discrepancy needs to be addressed.

3) All of the co-immunoprecipitation experiments are performed using over-expressed IRS4 and FER. The ability of endogenous proteins to interact should be assessed to validate this interaction.

4) In previous studies, the authors identified the MET receptor as a target of FER and showed that it plays a role in the ligand-independent activation of the receptor to promote invasion and migration. In the current study, inhibition of MET does not inhibit AKT activation indicating that these are two discreet pathways. It would be informative to determine if there is a mutually exclusive expression of IRS4 and MET in human ovarian tumors.

5) Throughout the manuscript, the authors state that FER-IRS4 promotes ovarian carcinoma proliferation. However, the assays show overall growth. The decrease in total cell numbers could reflect either proliferation or survival changes. Given the important role of PI3K-AKT signaling in survival, individual assays to assess the specific mechanism that contributes to growth should be performed.

6) The majority of experiments were performed using only OVCAR5 cells, with the exception that knockdown of IRS4 in OVCAR3 cells also suppressed growth. This limited use of cells raises concern about the general conclusions about IRS4 function in ovarian cancer. Inclusion of experiments using human PDX models of ovarian cancer would strengthen the clinical significance of the conclusions of the study. In addition, it would be informative to express IRS4 in other negative cell lines to determine its impact on their function.

*Reviewer #3:*

This is an interesting paper investigating the role of IRS4 as a decisive adapter protein in the growth of some ovarian cancers that links the FER tyrosine kinase to the PI3K/AKT by binding to PIK3R2. Generally, I like the paper because it combines cell-based molecular studies with the function of IRS4 to promote tumor growth; however, as I was reading the paper there were certain questions arising, which might be considered by the authors:

1. Regarding the interaction of FER with IRS4. The data suggest that the N-terminal region of IRS4 is important, which implicates the PH or PTB domain, whereas mutation of FER implicated the C-terminal part of the kinase for the interaction. Although the authors have all the tools, they leave this interaction question open for discussion. IRS PTB domains usually bind NPEpY motifs, which do not exist in FER; however, they discuss the possibility that a PGEpY motif plays this role. The authors should test this possibility. Is this site phosphorylated? It is also possible that the PH domain contributes to the interaction. PH domain in IRS tend to bind to negative peptide motifs. Do such motifs exist in FER and could that contribute to the binding specificity?

2. Regarding specificity, does FER interact with IRS1 or IRS2? By using HEK cells, the authors bias their analysis toward IRS4 as some clones of this line only express IRS4. Functional interactions with IRS1 and IRS2 might implicate this signaling system in metabolic or normal growth regulation. The authors should test this possibility in the assay systems. Does overexpression of IRS1 or 2 block IRS4 action? Otherwise, I am left with the feeling that FER-IRS4 is very specific---which might be the wrong impression.

3. Similar concerns arise for the specific role of PIK3R2 in the mechanism of Figure 6I. I would expect that IRS4 interacts with other PI3K regulatory isoforms. Are the results biased toward PIK3R2 owing to the selection of cells that over express this protein? If IRS4 does interact with other PIK regulatory isoforms, would such interactions also promote growth? Or would expression of the other isoforms suppress the formation of the FER/IRS4/PIK3R2 complex? I suspect they have the tools for this experiment.

4. What about the comparison of signaling generated by FER/IRS4 compared against IGF1R/IRS4, especially since IGF1R has been implicated in these systems previously. It seems the authors have the perfect system to investigate whether FER/IRS4 is fundamentally different than IGF1R/IRS4 at the growth regulatory level. Is IGF1R activated in the test cell system?

5. Immunostaining is of concern owing to the specificity of the IRS4 antibody. The authors should show that IRS4 antibody only recognizes IRS4 and not the other isoforms to validate their experiments in Figure 7.

[Editors’ note: further revisions were suggested prior to acceptance, as described below.]

Thank you for resubmitting your work entitled "FER-mediated phosphorylation and PIK3R2 recruitment on IRS4 promotes AKT activation and tumorigenesis in ovarian cancer" for further consideration by *eLife*. Your revised article has been evaluated by Jonathan Cooper (Senior Editor) and a Reviewing Editor.

This paper investigates the mechanism of cellular proliferation through FER (feline sarcoma-related kinase). The authors demonstrate that FER directly phosphorylates IRS4 (insulin receptor substrate 4) and that this tyrosine phosphorylation is important to create a binding site for recruiting the PI3K and activate the AKT cascade to promote proliferation. This can cause increased tumor burden in mice. This is important because ovarian cancer ranks 5th among all cancer related mortality in women.

The conclusion that FER / IRS4 contributes to tumor burden (Figure 7E) is generally consistent with the major conclusion that IRS4 signaling can exacerbate ovarian cancer burden. This is important; however, the relative contribution of IRS4 verses "the other" pathways is loosely established/discussed, especially as risk of death from ovarian cancer is still rather fatal in patients with low IRS4. The biochemical analysis is broadly consistent with the story, but several shortcomings weaken the experimental mechanism, which require attention by the authors.

Specific Points

1. The analysis of the interaction between IRS4 and FER is reasonably rigorous even though it relies largely upon overexpression of the relevant proteins; however, the analysis lacks identification of the binding sites in the PTK domain of FER that interact directly with the PH and PTB domains of IRS4. (lines 170- 84). Based upon previous work with IRS1, an NPXY motif might be expected as the PTB domain binding site. By contrast, a cluster of acid residues might be expected for the PH domain (See PMID 9813005). This should be established experimentally or at least discussed.

2. Line 221 suggests that there are at least five tyrosine phosphorylation sites in IRS1; however, some other tyrosine phosphorylation sites appear to exist, and this should be mentioned at line 222 (See Figure 3F). How might these sites contribute?

3. While the authors argue the PI3K-AKT-mTOR signaling pathway plays a critical role in controlling the proliferation of OVCAR-5 ovarian cancer cells, they do not clarify why the cells can grow without IRS4 and reduced AKT activity. Is the residual activity important or is there another pathway? Can the other pathway operate when IRS4 is not elevated in most other ovarian tumor cells (See Figure 4A and Figure 7E).

4. Although Tyr779 enhanced the recruitment of PIK3R2 and activation of the PI3K-AKT signaling pathway, is the sequence surrounding this site expected to be recognized by the SH2 domain? This point is important because PIK3R1 can complex with both WT and Y779F mutant of IRS4 which should have a similar motif specificity. These results suggest that an intermediate might be involved. There are many possibilities in the MS/MS list. Generally, the different contributions of PIK3R1 and PIK3R2 seem confusing and ignored by the authors.

5. Inclusion of pAKT levels in Figure 5H would strengthen the conclusion that Y779-dependent recruitment of PIK3R2 to IRS4 is primarily responsible for activating AKT to drive growth.

6. Tyrosine phosphorylation of IRS4 and the binding of PIK3R2 decreased upon FER-ko, but it seems substantial activity remains (See Figure 6C). Moreover, IGF1R inhibition failed to implicate the IGF1R as an alternative kinase, which is a reasonable alternative; however, perhaps IGF1 should have been added to the assay.

7. At line 358…. TAE684 is not a specific FER inhibitor, so the conclusion of the experiment, while consistent with the story, might be wrong.

8. At line 378…. It is hard to determine that the FER kinase mediated tyrosine phosphorylation of IRS4 plays a key function in controlling cell proliferation in ovarian cancer because the cells still grow, although slowly. It might be important to inactivate the other pathways to cause complete inhibition of growth, and then show that FER-> IRS4 can restore (rescue) tumor growth as demonstrated decades ago with 32D cells experiments (PMID: 8798677). The in vivo tumor growth experiments do show a contribution of FER Irs4 in tumor burden. Regardless, the absence of IRS4 slows but does not prevent tumor growth, so it is important to understand the alternative pathway(s), which also lead cause substantial patient death.

9. Line 442-444: The role of SHP2 in IRS1 and IRS2 has not been clearly demonstrated to attenuate PI3K signaling. Thus, whether the lack of SHP2 in IRS4 exacerbates PI3K signaling could be deleted unless an appropriate reference is cited.

10. Lines 450-452: The authors should estimate of the contribution of IRS4 signaling versus other pathways for tumor survival or burden in order to establish the therapeutic benefit for disruption of the FER-IRS4 pathway. Based on Figure 7E the improvement seems significant but small, so the other pathways must be rather robust. What are they?

11. Lines 454-461: Can the authors tell us which of the MS/MS verified Tyr(P) sites reside in YXXM motifs to create potential PI3K binding sites. This might be achieved from clearer writing.

12. Inclusion in the Discussion of how the current study differs from the previous knockout study of FER would clarify the discrepancies in the outcomes of the two studies. The authors provide this explanation in the response to reviewers and should include it in the Discussion.

13. On Page 10, line 270, LY294002 is described as an AKT inhibitor. This is incorrect, LY294002 is a PI3K inhibitor.

---

## [Author Response]

[Editors’ note: The authors appealed the original decision. What follows is the authors’ response to the first round of review.]

This is an interesting study that identified IRS4 as a mediator of FER signaling. However, a number of problems with thus study were identified by the reviewers. First, the study exhibits a reliance on the use of over-expression (rather than endogenous) protein analysis; second, the conclusions presented conflict with prior published work by the senior author (MET etc) and these conflicts are not explained; and third, the relevance of the conclusions to human cancer are not fully established. The specific issues noted by the reviewers are appended to this letter. It is clear that substantial revision of your manuscript will be required. It is the policy of eLife to return the manuscript to the authors under these circumstances.Reviewer #1:This is an interesting paper that demonstrates a role for IRS4 as a target of Fer signaling that leads to PIK3R2 recruitment and Akt activation that may be relevant to ovarian cancer. This pathway is demonstrated to promote the growth of human ovarian cancer cells in in vitro and in a xenograft assay.The major novelty of this study is the demonstration that this pathway is independent of Met/Gab1, which was previously implicated in Fer-stimulated AKT activation. This is a surprising finding. Otherwise, the study appears to be a relatively standard biochemical analysis with minimal attention to cancer relevance.There are some problems with the study presented that need to be addressed.a) All the biochemical studies appear to rely on over-expressed proteins. The study would be improved by showing that the reported protein interactions are detected in assays using endogenous proteins.

We thank the reviewer to point out this issue. To address this, we performed endogenous co-immunoprecipitation assay in OVCAR5 cell line, which has aberrantly high expression of both FER kinase (Ref 16) and IRS4 (Figure 4A). Compared to anti-IgG control, the OVCAR5 cell lysates with anti-IRS4 or anti-FER antibody showed the interaction between FER and IRS4 at endogenous level (Figure 6A, line 342-344).

Meanwhile, we also performed co-immunoprecipitation assay in single clone-derived IRS4 knockout or FER knockout OVCAR5 cell lines, and observed endogenous binding between FER and IRS4 in parental OVCAR5 cells as well (Figure 6B, line 344-345).

These results demonstrate the physical interaction between FER and IRS4 at endogenous level in OVCAR5 ovarian cancer cells.

b) What is the evidence that IRS4 is a Fer substrate? The data presented seems to be limited to a correlation between Fer over-expression and knockout with IRS4 tyrosine phosphorylation in cultured cells. Is IRS4 phosphorylated by Fer in vitro?

We thank the reviewer for this suggestion. To address this issue, we expressed and purified Myc-tagged human IRS4 protein in HEK293FT cells, and used this as potential substrate to perform in vitro *Kinase* (IVK) assay with purified GST-tagged human FER kinase (C-terminal tyrosine kinase domain 541-822aa). The tyrosine phosphorylation level of IRS4 was increased in a FER kinase dosage-dependent manner (Figure 3B, line 191-195). This result demonstrates that IRS4 can be phosphorylated by FER in vitro and is a bona fide FER substrate.

c) Some of the details of experimental procedures are unclear. For example, three different guide RNAs are described for Fer KO cells, yet only a single line of Fer KO cells appears to be presented. Which guide RNA was used? Similarly, three different guide RNAs are described for IRS4 KO lines and cell two lines were reported. Which guide RNAs were used?

We thank the reviewer to point out this confusion. We have clarified this missing information in our revised manuscript. For constructing FER KO cell line, guide RNA sgRNA#4 (5’-AGAGTTTGATACTTCCTTAC-3’) was used. For constructing IRS4 KO cell lines, guide RNAs sgRNA#1 (5’-CCATCGCGAAGTATTCGTCT-3’) and sgRNA#2 (5’-TATAGGGTGATCACGCGCCG-3’) were used (line 535-536).

d) The tumors in the xenograft study appear to be large (>2,000 m3). The authors should confirm that tumors this large are acceptable to the institutional animal care and use committee.

We thank the reviewer to point out this issue. We have confirmed with the institutional animal care and use committee in ShanghaiTech University. The tumor volume standard here is also less than 2,000 mm^3^.

Author response image 1 shows our original tumor volume data for day30 and day33 (Endpoint). As you can see, at Day30, tumor volume from all groups is less than 2,000 mm^3^. Therefore, we decided to acquire another data point three days later. However, at Day33, tumor volume of 3 mice from IRS4 WT rescue group exceeded 2,000 mm^3^. We euthanized all the mice right away.

**Author response image 1. sa2fig1:** 

In our revised manuscript, we will illustrate tumor volume data till Day30, as shown in Figure 6H, line 382-392.

Reviewer #2:Summary:This study examines the mechanism by which the FER non-receptor tyrosine kinase contributes to ovarian carcinoma cell growth. Mass spectrometry was used to identify targets of the FER kinase and IRS4 was identified as one of the top candidates. As reported by the authors, high IRS4 expression correlates with worse outcomes in ovarian cancer patients, which emphasizes the significance of this study. Overall the data presented are clear and support a role for IRS4 in growth regulation. However, the data are limited to primarily one cell line and over-expression studies which diminishes the impact. Moreover, the novel mechanistic information gained regarding IRS4 function is limited. For these reasons, publication in eLife is considered premature at this time.1) IRS4 belongs to the family of insulin receptor substrate proteins that have been best characterized for mediating signaling downstream of the insulin and insulin-like growth factor-1 (IGF-1) receptors to regulate organism metabolism and growth, respectively. IGF-1R was also identified as a target of FER in the mass-spec analysis. The authors do not address if the function of IRS4 is dependent upon this FER-mediated phosphorylation of IGF-1R. In addition, the IRS4 tyrosine mutations could also interfere with IGF-1R signaling to impact tumor growth, and this should be investigated to be able to attribute the IRS4-dependent effects to FER.

We thank the reviewer to point out this issue. To address if the function of IRS4 is dependent upon FER-mediated phosphorylation of IGF-1R, we applied IGF-1R inhibitor BMS-536924 (S1012) to inhibit IGF-1R activity in OVCAR5 cell line. The efficiency of BMS-536924 was pretty good, as demonstrated by decreased levels of pY1131 IGF-1R. However, pharmacological inhibition of IGF-1R didn’t decrease tyrosine phosphorylation of IRS4 or recruitment of PIK3R2 in OVCAR5 cells (Figure 6—figure supplement 1, line 352-355). In contrast, knockout of FER or treatment with FER inhibitor TAE684 (S1108) in OVCAR5 cells inhibited the phosphorylation of IRS4, as well as the recruitment of PIK3R2 (Figure 6C-D, line 347-352 and line 357-364).

These results suggest that the phosphorylation and activation of IRS4 has the IGF1R-independent pathway which mediated by kinase FER.

2) In a previous study (G. Fan et al., Genes and Dev, 2016), ovarian carcinoma cells with or without expression of FER were injected subcutaneously in mice and differences in tumor growth were not observed. However, metastasis to the lungs was decreased. In the current study, tumor growth was suppressed when IRS4 expression was knocked out. This raises concern that the growth-promoting effects of IRS4 are independent of FER. This discrepancy needs to be addressed.

We thank the reviewer to point out this issue. In this current manuscript, we have provided both in vitro (Figure 3B) and cell-based (Figure 3A and 3C-F) biochemical assays to demonstrate the kinase-substrate regulatory mode between FER and IRS4. Further loss-of-function assay also suggest that FER-mediated tyrosine phosphorylation of IRS4 is important for PIK3R2 recruitment (Figure 5 and Figure 6C-D) and downstream AKT pathway activation (Figure 4F, 4I and 6D), which is important for ovarian cancer cell proliferation (Figure 4 and 6).

There is one determining factor should be taken into consideration for solving abovementioned discrepancy. In the previous paper *(G. Fan et al., Genes & Dev, 2016)*, the majority of in vivo assays was done with CAOV4 cells, which has no IRS4 expression at all. In this current manuscript, two cell lines we used in our phenotypic assays, OVCAR5 and OVCAR3, have high IRS4 expression. We think this key difference could explain the different functional output of FER in both experimental context: In the absence of IRS4, FER regulates ovarian cancer cell motility and invasiveness mainly through MET-GAB1-SHP2-ERK1/2 signaling pathway, with MET and GAB1 as its substrates; in IRS4-positive ovarian cancer cells, FER-mediated phosphorylation of Tyr779 enables IRS4 to recruit PIK3R2/p85β, the regulatory subunit of PI3K, and activate the PI3K-AKT pathway for proliferation (line 431-440).

3) All of the co-immunoprecipitation experiments are performed using over-expressed IRS4 and FER. The ability of endogenous proteins to interact should be assessed to validate this interaction.

We thank the reviewer to point out this issue. To address this, we performed endogenous co-immunoprecipitation assay in OVCAR5 cell line, which has aberrantly high expression of both FER kinase (Ref 16) and IRS4 (Figure 4A). Compared to anti-IgG control, the OVCAR5 cell lysates with anti-IRS4 or anti-FER antibody showed the interaction between FER and IRS4 at endogenous level (Figure 6A, line 342-344).

Meanwhile, we also performed co-immunoprecipitation assay in single clone-derived IRS4 knockout or FER knockout OVCAR5 cell lines, and observed endogenous binding between FER and IRS4 in parental OVCAR5 cells as well (Figure 6B, line 344-345).

These results demonstrate the physical interaction between FER and IRS4 at endogenous level in OVCAR5 ovarian cancer cells.

4) In previous studies, the authors identified the MET receptor as a target of FER and showed that it plays a role in the ligand-independent activation of the receptor to promote invasion and migration. In the current study, inhibition of MET does not inhibit AKT activation indicating that these are two discreet pathways. It would be informative to determine if there is a mutually exclusive expression of IRS4 and MET in human ovarian tumors.

We thank the reviewer for this suggestion. To determine if there is a mutually exclusive expression of IRS4 and MET in human ovarian tumors, we performed (1) IHC analysis for both IRS4 and MET in ovarian tumor tissue samples; (2) TCGA RNA seq data analysis. We couldn’t get significant P-value from either analysis. However, we did observe slight anti-correlation trend, as shown in Author response image 2 in both analyses.

5) Throughout the manuscript, the authors state that FER-IRS4 promotes ovarian carcinoma proliferation. However, the assays show overall growth. The decrease in total cell numbers could reflect either proliferation or survival changes. Given the important role of PI3K-AKT signaling in survival, individual assays to assess the specific mechanism that contributes to growth should be performed.

We thank the reviewer for this great suggestion. To address this issue, we first performed Annexin V-FITC and Propidium Iodide (PI) double staining assay in both WT and IRS4 KO cell lines. Results indicated that loss of IRS4 has minimal impact on ovarian cancer cell survival (Figure 4—figure supplement 1A, line 250-253).

We further performed cell cycle analysis in both WT and IRS4 KO cell lines with Propidium Iodide (PI) staining. Results indicated that loss of IRS4 significantly decreased the proportion of cells within S phase (Figure 4—figure supplement 1B-C, line 253-255).

Combining together, these results highly suggest that FER-IRS4 promotes ovarian carcinoma proliferation rather than survival.

6) The majority of experiments were performed using only OVCAR5 cells, with the exception that knockdown of IRS4 in OVCAR3 cells also suppressed growth. This limited use of cells raises concern about the general conclusions about IRS4 function in ovarian cancer. Inclusion of experiments using human PDX models of ovarian cancer would strengthen the clinical significance of the conclusions of the study. In addition, it would be informative to express IRS4 in other negative cell lines to determine its impact on their function.

We thank the reviewer for this great suggestion. To address this issue, we stably expressed IRS4 into two negative cell lines (HEY and OVCAR8) to further determine its impact on ovarian carcinoma proliferation. Our results suggested that stably expressed IRS4: (1) elevated AKT kinase activation (Figure 4B and 4D, line 237-239 and 264-266); (2) increased ovarian cancer cell proliferation in vitro (Figure 4C and 4E, line 239-241); (3) accelerated tumor growth in vivo (Figure 6I-6L, line 392-393).

To be honest, human PDX model is challenging for our signaling transduction research-orientated lab. Hopefully the reviewer will be satisfied with the data we provide in two IRS4-negative cell lines.

Reviewer #3:This is an interesting paper investigating the role of IRS4 as a decisive adapter protein in the growth of some ovarian cancers that links the FER tyrosine kinase to the PI3K/AKT by binding to PIK3R2. Generally, I like the paper because it combines cell-based molecular studies with the function of IRS4 to promote tumor growth; however, as I was reading the paper there were certain questions arising, which might be considered by the authors:1. Regarding the interaction of FER with IRS4. The data suggest that the N-terminal region of IRS4 is important, which implicates the PH or PTB domain, whereas mutation of FER implicated the C-terminal part of the kinase for the interaction. Although the authors have all the tools, they leave this interaction question open for discussion. IRS PTB domains usually bind NPEpY motifs, which do not exist in FER; however, they discuss the possibility that a PGEpY motif plays this role. The authors should test this possibility. Is this site phosphorylated? It is also possible that the PH domain contributes to the interaction. PH domain in IRS tend to bind to negative peptide motifs. Do such motifs exist in FER and could that contribute to the binding specificity?

We thank the reviewer to give this great suggestion. In our previous manuscript, whereas the 335-400 and 401-1257 mutants demonstrated no binding affinity with FER, the 200-400 mutant maintained weak but substantial interaction with FER (Figure 2E). The N-terminal mutant 1-334 showed strongest binding among all these truncated constructs (Figure 2E, lane 153-154). Compared to the 200-400 mutant, which only covers PTB domain, the N-terminal 1-334aa of IRS4 contains both PH and PTB domains. To solve this issue, we constructed the 1-400 (both PH and PTB domains), 1-200 (PH domain only) and 200-400 (PTB domain only) mutants of IRS4 to further narrow down the binding region on IRS4. However, all three mutants showed as strong binding affinity with FER as WT IRS4, suggesting both PH and PTB domains were involved (Figure 2—figure supplement 2A, line 156-160). Meanwhile, we also constructed Myc-IRS4ΔPH and IRS4ΔPTB mutants to further dissect their individual roles in association with FER. Interestingly, deletion of PH or PTB domain has minimal effect on binding affinity with FER (Figure 2—figure supplement 2B, line 160-164), suggesting disrupt either of these domains is not sufficient to collapse the protein complex. Together, these results indicated that both PH and PTB domains of IRS4 are participated in the association with the kinase FER.

IRS4 PTB domains usually bind NPEpY motifs, which do not exist in FER; however, there is one PGEpY motif within FER protein sequence. We either mutated key tyrosine residue (Y492F) or deleted this motif completely (Δ489-492), and compared IRS4-binding affinity of these mutants with wt FER. However, compared to FER WT, these FER mutants showed equivalent binding affinity with IRS4, as shown in Author response image 3.

**Author response image 3. sa2fig3:** 

Meanwhile, we constructed GFP-FER 447-822 (SH2+kinase domains), GFP-FER 447-550 (SH2 domain) and GFP-FER 563-822 (kinase domain) truncation mutants to further narrow down the binding region on FER kinase. Notably, FER kinase domain, but not SH2 domain, is involved in the interaction with IRS4 (Figure 2G, line 171-177). Furthermore, FER kinase domain, but not SH2 domain, shows strong interaction with either PH domain (1-200aa) or PTB domain (200-400aa) of IRS4 (Figure 2—figure supplement 2C-D, line 177-179). Together, these results indicated that both PH and PTB domains of IRS4 are participated in the association with the kinase domain of FER.To further narrow down the binding region on FER kinase domain, we have made the following predictions and verifications. The PH domain is a functional domain present in a variety of signaling and cytoskeleton-related proteins. The polarity of the PH domain suggests that the ligand may be negatively charged. In our previous conclusion, both PH and PTB domains of IRS4 are participated in the association with the kinase domain of FER. Then we planned to analyze the charge distribution on the surface of FER kinase domain.

The crystal structure of the target protein was obtained from the AlphaFold Protein Structure Database (PDB database, https://www.alphafold.ebi.ac.uk/). To analyze the charge distribution on the protein surface, UCSF Chimera v1.14 (https://www.cgl.ucsf.edu/chimera/) was used to display the 3D structures and label the charged amino acids. Figures were generated using UCSF Chimera v 1.14.

As expected above, the surface of the PH domain in IRS4 is mainly distributed with positive charges. Surprisingly, there is a negative charge distribution on the surface of FER kinase domain, where E676, D684 and E740 are key amino acid residues. We then constructed single or multiple mutants of these key amino acids of FER, and detected their interaction with IRS4 full-length or 1-200 (PH) mutants in the overexpressed system, respectively, to verify whether the mutations of negative amino acids in the FER kinase domain would affect their binding with IRS4. The results suggested that the single or multiple mutations of negative amino acids in the FER kinase domain failed to disrupt their interaction with WT IRS4 or 1-200 (PH) mutants of IRS4.

**Author response image 4. sa2fig4:** 

**Author response image 5. sa2fig5:** 

Combining together, these results highly suggest both PH and PTB domains of IRS4 are participated in the association with the kinase domain of FER (Figure 2H).

2. Regarding specificity, does FER interact with IRS1 or IRS2? By using HEK cells, the authors bias their analysis toward IRS4 as some clones of this line only express IRS4. Functional interactions with IRS1 and IRS2 might implicate this signaling system in metabolic or normal growth regulation. The authors should test this possibility in the assay systems. Does overexpression of IRS1 or 2 block IRS4 action? Otherwise, I am left with the feeling that FER-IRS4 is very specific---which might be the wrong impression.

We thank the reviewer to give this great suggestion. Actually, Masanori Iwanishi and his colleagues have showed interaction between IRS1 and FER at endogenous level in 3T3-L1 adipocytes (JBC, 2000). We also performed endogenous co-immunoprecipitation assay in OVCAR5 cell line. Compared to anti-IgG control, the OVCAR5 cell lysates with anti-FER antibody showed the interaction between FER and IRS1 at endogenous level (Figure 2—figure supplement 1A, line 136-138).

Then we constructed IRS1 and IRS4 plasmids, and expressed these constructs alone or in combination, as indicated, in HEK293 cells. We confirmed strong binding between IRS4 and FER. Notably, the binding affinity was not affected in the absence and/or presence of IRS1 (Figure 2—figure supplement 1B, line 138-140).

3. Similar concerns arise for the specific role of PIK3R2 in the mechanism of Figure 6I. I would expect that IRS4 interacts with other PI3K regulatory isoforms. Are the results biased toward PIK3R2 owing to the selection of cells that over express this protein? If IRS4 does interact with other PIK regulatory isoforms, would such interactions also promote growth? Or would expression of the other isoforms suppress the formation of the FER/IRS4/PIK3R2 complex? I suspect they have the tools for this experiment.

We thank the reviewer to give this great suggestion. We first constructed PIK3R1 and PIK3R2 plasmids, and expressed these constructs alone or in combination, as indicated, in HEK293 cells. We confirmed interaction between IRS4 and PIK3R2. Notably, the binding affinity was not affected in the absence and/or presence of PIK3R1 (Figure 5—figure supplement 1A , line 324-328). These data suggest that expression of the other isoforms of PI3K wouldn’t suppress the formation of the IRS4-PIK3R2 complex.

In our original manuscript, we showed that phoshphorylation of Tyr779 on IRS4 is important for PIK3R2 recruitment (figure 5B-5H). Next, we want to test whether IRS4 could recruit PIK3R1 through same mechanism. Interestingly, our data indicated that PIK3R1 could form complex with both WT and Y779F mutant of IRS4 with equivalent binding affinity (Figure 5—figure supplement 1B, line 328-330). Therefore, both PIK3R1 and PIK3R2 can be recruited to IRS4, but with distinctive binding mechanism.

Notably, PIK3R1/p85α has a tumor-suppressor function, whereas PIK3R2/p85β is an oncogene. Expression of PIK3R2, encoding the p85β regulatory subunit of PI3K, increases with advanced tumor stage in melanoma, breast, and squamous cell lung carcinoma. Its overexpression induces metastasis in mouse models, whereas preclinical deletion of PIK3R2 triggers tumor regression and reduces invasion. Therefore, the abovementioned difference in binding mechanism with IRS4 may shed some light to decipher their opposite roles in tumorigenesis.

4. What about the comparison of signaling generated by FER/IRS4 compared against IGF1R/IRS4, especially since IGF1R has been implicated in these systems previously. It seems the authors have the perfect system to investigate whether FER/IRS4 is fundamentally different than IGF1R/IRS4 at the growth regulatory level. Is IGF1R activated in the test cell system?

We thank the reviewer to point out this issue. To address if the function of IRS4 is dependent upon FER-mediated phosphorylation of IGF-1R, we applied IGF-1R inhibitor BMS-536924 (S1012) to inhibit IGF-1R activity in OVCAR5 cell line. The efficiency of BMS-536924 was pretty good, as demonstrated by decreased levels of pY1131 IGF-1R. However, pharmacological inhibition of IGF-1R didn’t decrease tyrosine phosphorylation of IRS4 or recruitment of PIK3R2 in OVCAR5 cells (Figure 6—figure supplement 1, line 352-355). In contrast, knockout of FER or treatment with FER inhibitor TAE684 (S1108) in OVCAR5 cells inhibited the phosphorylation of IRS4, as well as the recruitment of PIK3R2 (Figure 6C-D, line 347-352 and 357-364).

These results suggest that the phosphorylation and activation of IRS4 has the IGF1R-independent pathway which mediated by kinase FER.

5. Immunostaining is of concern owing to the specificity of the IRS4 antibody. The authors should show that IRS4 antibody only recognizes IRS4 and not the other isoforms to validate their experiments in Figure 7.

We thank the reviewer to point out this issue. In this current manuscript, we have adopted a xenograft mouse model with subcutaneous injection of OVCAR-5 IRS4-KO cells which was rescued with either an empty vector, or WT IRS4. Compared to mice injected with OVCAR-5 IRS4-KO cells rescued with WT IRS4, we observed significantly delayed tumor formation in mice injected with OVCAR-5 IRS4-KO cells with an empty vector (Figure 6H).

Most importantly, we collected 2 xenograft tumor sample derived from OVCAR-5 IRS4-KO cells rescued with an empty vector or WT IRS4, respectively, to further explore the specificity of IRS4 antibody by immunohistochemistry staining. The results showed that IRS4 antibody was quite specific for IRS4, since no signal was observed in xenograft tumor sample derived from IRS4-KO OVCAR5 cells rescued with an empty vector (Figure 7—figure supplement 1, line 407-409).

In addition, we constructed FER, IRS4 and IRS1 plasmids, and expressed them in HEK293 cells. We then tested whether or not IRS4 antibody can only detect IRS4 protein. The result suggests that IRS4 antibody used in this manuscript is specific and couldn’t recognize other IRS isoforms.

**Author response image 6. sa2fig6:** 

Finally, we used ENDscript website (https://espript.ibcp.fr/ESPript/cgi-bin/ESPript.cgi) to perform sequence alignments among IRS4, IRS1 and IRS2, as shown in Author response image 7 and Author response image 8. The green shaded areas represent the recognition site for anti-IRS4 antibody. The results indicated that IRS4 has low sequence identity and/or similarity with either IRS1 or IRS2, especially in the epitope region of anti-IRS4 antibody we used in the manuscript.

**Author response image 7. sa2fig7:** Protein sequence alignment between IRS4 and IRS1.</Author response image 7 title/legend>.

**Author response image 8. sa2fig8:** Protein sequence alignment between IRS4 and IRS1.

[Editors’ note: what follows is the authors’ response to the second round of review.]

This paper investigates the mechanism of cellular proliferation through FER (feline sarcoma-related kinase). The authors demonstrate that FER directly phosphorylates IRS4 (insulin receptor substrate 4) and that this tyrosine phosphorylation is important to create a binding site for recruiting the PI3K and activate the AKT cascade to promote proliferation. This can cause increased tumor burden in mice. This is important because ovarian cancer ranks 5th among all cancer related mortality in women.The conclusion that FER / IRS4 contributes to tumor burden (Figure 7E) is generally consistent with the major conclusion that IRS4 signaling can exacerbate ovarian cancer burden. This is important; however, the relative contribution of IRS4 verses "the other" pathways is loosely established/discussed, especially as risk of death from ovarian cancer is still rather fatal in patients with low IRS4. The biochemical analysis is broadly consistent with the story, but several shortcomings weaken the experimental mechanism, which require attention by the authors.Specific Points1. The analysis of the interaction between IRS4 and FER is reasonably rigorous even though it relies largely upon overexpression of the relevant proteins; however, the analysis lacks identification of the binding sites in the PTK domain of FER that interact directly with the PH and PTB domains of IRS4. (lines 170- 84). Based upon previous work with IRS1, an NPXY motif might be expected as the PTB domain binding site. By contrast, a cluster of acid residues might be expected for the PH domain (See PMID 9813005). This should be established experimentally or at least discussed.

We thank the reviewer for this great suggestion. The pleckstrin homology (PH) domain is a functional domain present in a variety of signaling and cytoskeleton-related proteins. The polarity of the PH domain suggests that the ligand may be negatively charged. In our previous conclusion, the PH domain of IRS4 also participates in the association with the kinase domain of FER. Then we planned to analyze the charge distribution on the surface of FER kinase domain.

First, we obtain the crystal structure of the target protein from the AlphaFold Protein Structure Database (PDB database, https://www.alphafold.ebi.ac.uk/). Then, we used UCSF Chimera v1.14 (https://www.cgl.ucsf.edu/chimera/) to display the 3D structures and label the charged amino acids. Figures were generated using UCSF Chimera v 1.14.

As expected, the surface of the PH domain in IRS4 is mainly distributed with positive charges. Interestingly, there is a negative charge distribution on the surface of FER kinase domain, where E676, D684 and E740 are key amino acid residues. We constructed single or multiple mutants of these key amino acids of FER, and detected their interaction with IRS4 full-length or 1-200 (PH) mutants, respectively, to verify whether the mutations of negative amino acids in the FER kinase domain would affect their binding with IRS4. The results suggested that the single or multiple mutations of negative amino acids in the FER kinase domain failed to disrupt their interaction with WT IRS4 or 1-200 (PH) mutants of IRS4. These new data have been integrated as new Figure 2 —figure supplement 3A-E, the Results section lines 181-206, Author response image 4 and Author response image 5.

The phosphotyrosine binding (PTB) domain recognizes phosphotyrosine-containing motifs for protein-protein interaction. Based upon previous work with IRS1, an NPXY motif might be expected as the PTB domain binding site (PMID: 7499194). However, FER possesses no NPXY motif but a QPVY motif within its kinase domain. To test the necessity of this motif in binding with IRS4, we took two strategies by either mutating the key tyrosine residue (Y634F) or deleting this motif (Δ631-634) completely. However, compared to FER wt, these FER mutants showed an equivalent binding affinity with IRS4, as shown in new Figure 2 —figure supplement 3F.

Combining together, our current data suggest that both PH and PTB domains of IRS4 participate in the association with the kinase domain of FER (Figure 2H).

2. Line 221 suggests that there are at least five tyrosine phosphorylation sites in IRS1; however, some other tyrosine phosphorylation sites appear to exist, and this should be mentioned at line 222 (See Figure 3F). How might these sites contribute?

We thank the reviewer for this great suggestion. In our revised manuscript, we have mentioned that some other tyrosine phosphorylation sites may still exist.

To determine which region on IRS4 can be phosphorylated by FER, we overexpressed several Myc-tagged IRS4 truncation mutants in conjunction with FER in HEK293FT cells and demonstrated tyrosine residues between 700 and 1093 amino acids of IRS4 were potential substrate(s) for FER kinase. However, it also should not be ignored that the 1-550aa truncation mutant of IRS4 also possessed weak but detectable phosphorylation signals in the presence of FER kinase. As shown in Author response image 9, Tyrosine 487 of hIRS4 also resembles YXXM motif that upon phosphorylation is predicted to bind SH2 domains in the p85 regulatory subunit of PI3K, resulting in activation of p110 catalytic subunit. This new information has been integrated into the Discussion section lines 521-526.

**Author response image 9. sa2fig9:** 

3. While the authors argue the PI3K-AKT-mTOR signaling pathway plays a critical role in controlling the proliferation of OVCAR-5 ovarian cancer cells, they do not clarify why the cells can grow without IRS4 and reduced AKT activity. Is the residual activity important or is there another pathway? Can the other pathway operate when IRS4 is not elevated in most other ovarian tumor cells (See Figure 4A and Figure 7E).

We thank the reviewer for pointing out this issue. By using RNA interference technology (RNAi), Eva Cuevas and her colleagues have shown that IRS-4 plays an important role in HepG2 proliferation/differentiation and exerts its actions through ERK and p70S6K activation in a Ras/Raf/MEK1/2- and PI3Kinase/AKT-independent manner and in a PKC-dependent way (J Hepatol. 2007 PMID: 17408801).

Interestingly in our study, whereas there was a dramatic decrease of phospho-AKT signal upon IRS4 deletion, we also observed evident elevation in phospho-ERK level in O5 IRS4-KO cells (Figure 4F), indicating a compensatory effect of ERK signaling pathway which may contribute to the survival and proliferative capacities of OVCAR-5 cancer cells.

To test this hypothesis, we treated O5 WT cells with PI3K inhibitor LY294002 along with ERK inhibitor U0126, followed by CTG assay (new Figure 4M). The inhibition of both PI3K-AKT pathway and ERK pathway in O5 cells almost blocked cell growth completely. Therefore, in the absence of IRS4-mediated PI3K-AKT activation, OVCAR-5 ovarian cancer cells would upregulate ERK activity in a compensating manner, thereby enhancing survival and proliferative capacities. Our results highly suggested that targeting both PI3K-AKT and ERK pathways simultaneously will be an effective strategy to treat ovarian cancer. This new data has been integrated into the Results section lines 312-321.

4. Although Tyr779 enhanced the recruitment of PIK3R2 and activation of the PI3K-AKT signaling pathway, is the sequence surrounding this site expected to be recognized by the SH2 domain? This point is important because PIK3R1 can complex with both WT and Y779F mutant of IRS4 which should have a similar motif specificity. These results suggest that an intermediate might be involved. There are many possibilities in the MS/MS list. Generally, the different contributions of PIK3R1 and PIK3R2 seem confusing and ignored by the authors.

We thank the reviewer for pointing out this issue. There are 7 YXXM motifs on IRS4, which have been speculated as potential binding sites for the regulatory subunit of PI3K, including PIK3R1 and PIK3R2. Our results from mass spectrometry and site-directed mutagenesis analysis revealed 5 major tyrosine residues phosphorylated by FER kinase: Tyr-656, -779, -828, -847 and -921. Among them, Tyr-779, -828 and -921 reside in YXXM motifs to create potential PI3K binding sites.

Therefore, the answer is “Yes” that the sequence surrounding Tyr779 site is expected to be recognized by the SH2 domain of PI3K. Given the different outcomes of phos-Tyr779 in recruiting PIK3R2 and PIK3R1 onto IRS4, it is possible that an intermediate might be involved in differentiating the binding mechanisms of these two regulatory subunits of PI3K, as suggested by the reviewer. We will further digest the MS/MS list and test this hypothesis. What is also possible is that PIK3R1 and PIK3R2 may rely on different YXXM motifs for docking onto IRS4 and subsequent downstream signaling regulation. In this current study, we identified Tyr779 of IRS4 subjected to FER kinase-mediated phosphorylation. Furthermore, using a proximity-based tagging system, we determined that FER-mediated phosphorylation of Tyr779 enables IRS4 to recruit PIK3R2/p85β, the regulatory subunit of PI3K, and activate the PI3K-AKT pathway. The abovementioned two hypotheses definitely deserve more investigation as the continuous study of this project in the lab.

5. Inclusion of pAKT levels in Figure 5H would strengthen the conclusion that Y779-dependent recruitment of PIK3R2 to IRS4 is primarily responsible for activating AKT to drive growth.

We thank the reviewer for this great suggestion. In our revised manuscript, we have included pAKT levels in new Figure 5H.

6. Tyrosine phosphorylation of IRS4 and the binding of PIK3R2 decreased upon FER-ko, but it seems substantial activity remains (See Figure 6C). Moreover, IGF1R inhibition failed to implicate the IGF1R as an alternative kinase, which is a reasonable alternative; however, perhaps IGF1 should have been added to the assay.

We thank the reviewer for this great suggestion. In our manuscript, CRISPR-Cas9 mediated FER knockout in OVCAR-5 ovarian cancer cells or pharmacological inhibition of FER with TAE684 decreased tyrosine phosphorylation of IRS4 (Figure 6C-D). IGF1R indeed phosphorylated IRS4 in the presence of ligand IGF1 (new Figure 6—figure supplement 1A-B). However, small molecular IGF-1R inhibitor BMS-536924 didn’t decrease tyrosine phosphorylation of IRS4 in the absence of IGF1 (new Figure 6—figure supplement 1A-B), indicating FER-mediated phosphorylation and activation of IRS4 in an IGF1-independent manner. Moreover, IGF-1R phosphorylated IRS4 probably in a Tyr779-independent manner, since mutating Tyrosine 779 to Phenylalanine didn’t decrease phosphorylation level of IRS4 mediated by IGF1R receptor tyrosine kinase (new Figure 6—figure supplement 1C). These new data have been integrated into the Results section lines 389-396.

7. At line 358…. TAE684 is not a specific FER inhibitor, so the conclusion of the experiment, while consistent with the story, might be wrong.

We thank the reviewer for pointing out this issue. Due to the long-term lack of necessary attention of the importance of FER protein kinase, currently we not only know little about the activation mechanism and substrate of FER, but also the research and development of FER-specific inhibitors lags far behind other kinase proteins. Fortunately, Sabine Hellwig and his colleagues have identified TAE684 as a small molecule compound against FES tyrosine kinase (Cell Chem Bio. 2012 PMID: 22520759). Given that FES and FER kinase proteins belong to the same sub-family of non-receptor tyrosine kinases, we started to test whether TAE684 could be used as a small molecule kinase inhibitor against FER.

First, we measured FER kinase activity inhibition by TAE684. We overexpressed FER in HEK293FT cells, followed by TAE684 treatment. We used Tyr402 auto-phosphorylation signal as a readout for measurement of FER kinase inhibition. Results have shown that TAE684 can robustly inhibit the auto-phosphorylation of FER at Tyr402, with IC_50_ reaching 8.8 nM (new Figure 6—figure supplement 2A).

Second, we collected ovarian cancer cell lysates from OVCAR-5, OVCAR-3, OVCAR-8, and HEY, as well as myeloid leukemia cell lysates from HL-60 (as positive control) to check the expression of FES kinase. The results clearly demonstrated that there is no FES expression in ovarian cancer cell lines used in this study (new Figure 6—figure supplement 2B).

Third, along with RNAi analysis, Yiyan Zheng and his colleague have successfully used TAE684 as a small molecule kinase inhibitor against FER to further investigate the kinase function in cytoskeletal remodeling and potential ovarian cancer treatment (Nature Communications 2018, PMID: 29396402). Similar to that study, we also performed both genetic and pharmacological research side-by-side to rule out potential off-target effect(s) of the kinase inhibitor. These new data have been integrated into the Results section lines 400-406.

8. At line 378…. It is hard to determine that the FER kinase mediated tyrosine phosphorylation of IRS4 plays a key function in controlling cell proliferation in ovarian cancer because the cells still grow, although slowly. It might be important to inactivate the other pathways to cause complete inhibition of growth, and then show that FER-> IRS4 can restore (rescue) tumor growth as demonstrated decades ago with 32D cells experiments (PMID: 8798677). The in vivo tumor growth experiments do show a contribution of FER Irs4 in tumor burden. Regardless, the absence of IRS4 slows but does not prevent tumor growth, so it is important to understand the alternative pathway(s), which also lead cause substantial patient death.

We thank the reviewer for pointing out this issue. We have read through the reference recommended by the reviewer, and carefully planned another in vitro cell proliferation rescue experiment.

In question 3, we found that in the absence of IRS4-mediated PI3K-AKT activation, OVCAR-5 ovarian cancer cells would upregulate ERK activity in a compensating manner, thereby enhancing survival and proliferative capacities. Pharmacological inhibition of both PI3K-AKT pathway and ERK pathway in OVCAR-5 cells almost blocked cell growth completely. Therefore, ERK pathway is the alternative pathway we are looking for.

For this question, we first treated IRS4 knockout OVCAR-5 cells with ERK pathway inhibitor U0126, and observed profound inhibition in cell growth. Most importantly, re-expression of WT IRS4 in IRS4-KO cells exposed to U0126 treatment recovered the growth capacity to the same level as U0126 treatment alone, whereas 5YF mutant of IRS4 failed to rescue cell proliferation (new Figure 6F). This rescue experiment suggested that tyrosine phosphorylation of IRS4, executed by FER kinase and alternatively IGF-1R in the presence of ligand IGF-1, plays a significant role in controlling cell proliferation in ovarian cancer. Simultaneously targeting both PI3K-AKT and ERK pathways will be an effective strategy to treat ovarian cancer. This new data has been integrated into the Results section lines 423-428.

9. Line 442-444: The role of SHP2 in IRS1 and IRS2 has not been clearly demonstrated to attenuate PI3K signaling. Thus, whether the lack of SHP2 in IRS4 exacerbates PI3K signaling could be deleted unless an appropriate reference is cited.

We thank the reviewer for pointing out this issue and feel sorry for any unclarity in the original manuscript.

In order to investigate the function of the two SHP2 binding COOH-terminal tyrosines of IRS1, Morris White and his colleagues replaced them with phenylalanine. This mutant form of IRS1 failed to bind SHP2, and exhibited increased tyrosine phosphorylation, phosphatidylinositol 3′-kinase binding and activation of protein synthesis in response to insulin. These results clearly suggest that SHP2 attenuates the phosphorylation and downstream signal transmission of IRS1 and that the interaction of IRS1 and SHP2 is an important regulatory event which attenuates insulin metabolic responses (JBC 1998, PMID: 9756938).

By using liver-specific SHP2 KO mice, Fawaz Haj and his colleagues were able to show that SHP2 is a negative regulator of hepatic insulin action, and its deletion enhances the activation of PI3K/AKT pathway downstream of the insulin receptor (JBC 2010, PMID: 20841350).

However, no significant association was detected between IRS4 and SHP2 (JBC 1998, PMID: 9553137). John Hilkens and his colleagues have demonstrated that IRS4 expression in mammary epithelial cells induces constitutive PI3K/AKT pathway hyperactivation, insulin/IGF1-independent cell proliferation, anchorage-independent growth and in vivo tumorigenesis. The constitutive PI3K/AKT pathway hyperactivation by IRS4 is unique to the IRS family and the authors identified the lack of a SHP2-binding domain in IRS4 as the molecular basis of this feature (Nature Comm. 2016, PMID: 27876799).

In our revised manuscript, we have rephrased the sentence and cited these references properly (Discussion section lines 494-504).

10. Lines 450-452: The authors should estimate of the contribution of IRS4 signaling versus other pathways for tumor survival or burden in order to establish the therapeutic benefit for disruption of the FER-IRS4 pathway. Based on Figure 7E the improvement seems significant but small, so the other pathways must be rather robust. What are they?

We thank the reviewer for this great suggestion. We analyzed clinical data from over 600 ovarian cancer patients (http://www.kmplot.com) and plotted the overall survival curves for both the IRS4-high and the IRS4-low cohorts. The majority of ovarian cancer patient samples that are collected and deposited into kmplot database are grouped in the advanced disease stage. That may partially explain the improvement seems significant but small based on IRS4 expression and signaling.

In question 3, we found that in the absence of IRS4-mediated PI3K-AKT activation, OVCAR-5 ovarian cancer cells would upregulate ERK activity in a compensating manner, thereby enhancing survival and proliferative capacities. Pharmacological inhibition of both PI3K-AKT pathway and ERK pathway in OVCAR-5 cells almost blocked cell growth completely. Therefore, ERK pathway is the alternative pathway we are looking for.

In question 8, we treated IRS4 knockout OVCAR-5 cells with ERK pathway inhibitor U0126, and observed profound inhibition in cell growth. Most importantly, re-expression of WT IRS4 in IRS4-KO cells exposed to U0126 treatment recovered the growth capacity to the same level as U0126 treatment alone, whereas 5YF mutant of IRS4 failed to rescue cell proliferation. This rescue experiment suggested that tyrosine phosphorylation of IRS4, executed by FER kinase and alternatively IGF-1R in the presence of ligand IGF-1, plays a significant role in controlling cell proliferation in ovarian cancer.

Therefore, in sum, simultaneously targeting both IRS4-mediated PI3K-AKT and ERK pathways may deliver a more effective strategy to treat ovarian cancer (Discussion section lines 513-515).

11. Lines 454-461: Can the authors tell us which of the MS/MS verified Tyr(P) sites reside in YXXM motifs to create potential PI3K binding sites. This might be achieved from clearer writing.

We thank the reviewer for this great suggestion. There are 7 YXXM motifs on IRS4, which have been speculated as potential binding sites for the regulatory subunit of PI3K. Our results from mass spectrometry and site-directed mutagenesis analysis verified 3 Tyr(P) sites reside in YXXM motifs to create potential PI3K binding sites: Tyr-779, -828 and -921. This new information has been integrated into the Discussion section lines 517-525.

12. Inclusion in the Discussion of how the current study differs from the previous knockout study of FER would clarify the discrepancies in the outcomes of the two studies. The authors provide this explanation in the response to reviewers and should include it in the Discussion.

We thank the reviewer to point out this issue. In this current manuscript, we have provided both in vitro (Figure 3B) and cell-based (Figure 3A and 3C-F) biochemical assays to demonstrate the kinase-substrate regulatory mode between FER and IRS4. Further loss-of-function assay also suggests that FER-mediated tyrosine phosphorylation of IRS4 is important for PIK3R2 recruitment (Figure 5 and Figure 6C-D) and downstream AKT pathway activation (Figure 4F, 4I and 6D), which is important for ovarian cancer cell proliferation (Figure 4 and 6).

In a previous study (Genes Dev. 2016, PMID: 27401557), ovarian carcinoma cells with or without expression of FER were injected subcutaneously in mice, and differences in tumor growth were not observed. However, metastasis to the lungs was decreased. In the current study, tumor growth was suppressed when IRS4 expression was knocked out. There is one determining factor that should be taken into consideration for solving the abovementioned discrepancy.

In the previous paper, the majority of in vivo assays were done with CAOV4 cells, which have no IRS4 expression at all. In this current manuscript, two cell lines we used in our phenotypic assays, OVCAR-5 and OVCAR-3, have high IRS4 expression. We think this key difference could explain the different functional output of FER in both experimental contexts: In the absence of IRS4, FER regulates ovarian cancer cell motility and invasiveness mainly through MET-GAB1-SHP2-ERK1/2 signaling pathway, with MET and GAB1 as its substrates; in IRS4-positive ovarian cancer cells, FER-mediated phosphorylation of Tyr779 enables IRS4 to recruit PIK3R2/p85β, the regulatory subunit of PI3K, and activate the PI3K-AKT pathway for proliferation. This new information has been integrated into the Discussion section lines 531-545.

13. On Page 10, line 270, LY294002 is described as an AKT inhibitor. This is incorrect, LY294002 is a PI3K inhibitor.

We thank the reviewer for pointing out this issue. Indeed, LY294002 is the first synthetic small molecule known to inhibit PI3Kα/δ/β with IC50 of 0.5 μM/0.57 μM/0.97 μM, respectively, in cell-free assays. LY294002 inactivates AKT/PKB, thus inhibiting cell proliferation and inducing apoptosis. We have corrected all the mistakes in our revised manuscript.